# CAUSAL PIECES: ANALYSING AND IMPROVING SPIKING NEURAL NETWORKS PIECE BY PIECE

## ABSTRACT

We introduce *causal pieces*, a novel concept for analysing spiking neural networks (SNNs), inspired by *linear pieces* used to study expressivity and trainability in artificial neural networks (ANNs). Causal pieces partition the input and parameter space of an SNN into distinct regions where the same subnetwork causes the output spikes of the SNN. For networks of integrate-and-fire neurons with exponential synapses, we show that within each causal piece, output spike times are locally Lipschitz continuous with respect to the input spike times and network parameters. We also prove that the number of causal pieces is a measure of the approximation capabilities of SNNs. Empirically, we find that parameter initialisations yielding more causal pieces on the training set strongly correlate with SNN training success. Remarkably, even SNNs with only positive weights can exhibit a high number of causal pieces, allowing them to achieve competitive performance on diverse benchmarks such as Yin-Yang, MNIST, and EuroSAT, compared to fully-connected ANNs. These results establish causal pieces as a powerful and principled tool for analysing and improving the computational capabilities of SNNs.

## 1 INTRODUCTION

Spiking neural networks (SNNs) have recently received increased attention due to their ability to facilitate low-power hardware solutions for deep learning methods, particularly for edge applications, e.g., in outer space onboard spacecraft (Izzo et al., 2022; Schumann, 2022; Lunghi et al., 2024). In large parts, this is caused by the development of methods and software tools that allow the usage of error backpropagation to train SNNs (Neftci et al., 2019; Mostafa, 2017; Göltz et al., 2021; Comsa et al., 2020; Klos & Memmesheimer, 2025), as well as emerging spike-based hardware systems (Frenkel et al., 2023) such as Intel's digital Loihi (Davies et al., 2018; Orchard et al., 2021) and the analog BrainScaleS-2 (Cramer et al., 2022; Spilger et al., 2023) chip, which promise not only low energy footprints, but accelerated computation. However, even though SNNs have been introduced already decades ago (Maass, 1994; 1997), their computational capabilities remain poorly understood Singh et al. (2023); Neuman et al. (2024), and it is still an open question whether spike-based neurons have any relevant benefit compared to their non-spiking counterparts commonly used in deep learning (Davidson & Furber, 2021; Yin et al., 2021; Kucik & Meoni, 2021; Lunghi et al., 2024; Dampfhoffer et al., 2022).

In this work, we introduce a framework for characterising SNNs inspired by *linear pieces*, a concept used to analyse the expressivity of ReLU-based ANNs (Frenzen et al., 2010; Montufar et al., 2014; Telgarsky, 2016; Hanin & Rolnick, 2019). Our main contribution is the idea of *causal pieces*: a structured way to partition the input and parameter space of an SNN into regions where output spikes are caused by the same subnetwork. Crucially, we show that the number of causal pieces provides a lower bound on the approximation error of SNNs: networks with more pieces are capable of modelling data better. In contrast to prior theoretical work, which is often limited to simplified neuron models or requires that the SNN is a continuous function of its inputs and parameters (Stanojevic et al., 2023; Zador & Pearlmutter, 1996; Maass & Schmitt, 1999; Neuman et al., 2024; Singh et al., 2023), we provide results that are exact for networks of integrate-and-fire (IF) neurons with exponential synapses – a special case of the widely used leaky integrate-and-fire (LIF) neuron model – even if spike times can change discontinuously.

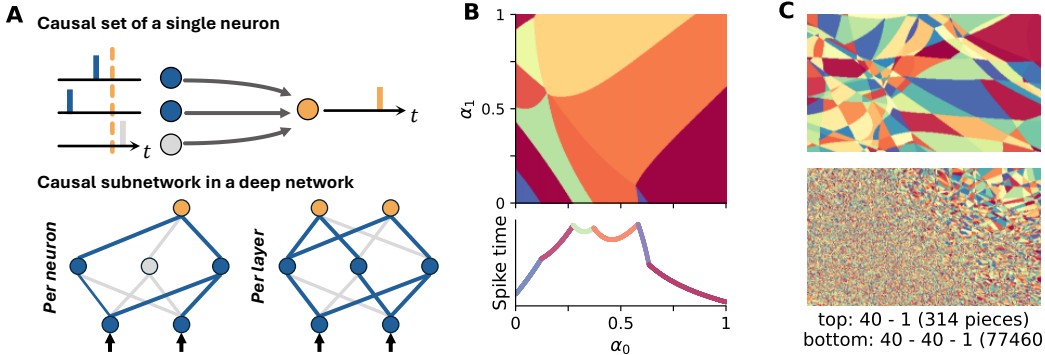

Figure 1: Causal sets and causal pieces. **(A)** Top: A single output neuron (orange) receiving input from three neurons. Only those input neurons (blue) that spike before the output neuron (i.e., before the dotted line) are part of the causal set. Bottom: In deep networks, this corresponds to subnetworks (blue), here shown for a single output neuron (left), or the whole output layer (right). **(B)** Top: Illustration of causal pieces of a single neuron. Bottom: The output spike time of the neuron when moving along the x-axis. Although the subnetwork remains fixed across a causal piece, the output spike time changes. **(C)** Causal pieces of the output neuron for two networks with different depth.

Formally, a causal piece is a subset of the inputs and network parameters where the output spikes of an SNN are always caused by the same subnetwork, meaning that the same subset of neurons and synapses in the SNN determine its output. For a single output neuron in a perceptron, this includes all input neurons with spike times preceding the output spike (Fig. 1A, top). In deep networks, causal pieces correspond to the subnetworks responsible for the spike times of individual output neurons or the entire output layer (Fig. 1A, bottom).

In Fig. 1B, we illustrate how the input domain of an SNN is decomposed into causal pieces (coloured regions, network parameters are kept fixed for simplicity). Within each piece, the output spike times of the SNN are not constant. Instead, the SNN can realise different non-linear functions in each piece, overall making it a piecewise continuous, non-linear function of the input (Fig. 1B, bottom). Although the number of pieces is determined by the network parameters, a simple way of increasing it is by adding more layers to the SNN (Fig. 1C).

Specifically, the contribution of our work is as follows:

1. We introduce the concept of causal pieces for SNNs and provide methods to count them.

2. Based on the proof for linear pieces (Frenzen et al., 2010), we prove – for IF neurons with exponential synapses – that the number of causal pieces is a measure of expressivity.

3. In simulations, we show that the number of causal pieces the training data fall into at network initialisation strongly correlates with training success (Fig. 3), providing a principled approach to guide SNN initialisation currently missing in the literature (Rossbroich et al., 2022). We show this for the Yin-Yang dataset (Kriener et al., 2022), Fashion-MNIST Xiao et al. (2017), and EuroSAT (Helber et al., 2019).

4. Furthermore, we see that hidden layers tend to increase the number of causal pieces, with the greatest benefit coming from initial layers (Fig. 5).

5. Lastly, we find that SNNs with only positive weights have a remarkably high number of causal pieces (Fig. 7), allowing them to reach typical performance levels of fully connected ANNs in standard benchmarks such as Yin-Yang, MNIST (LeCun et al., 2010), and EuroSAT.

In the following, we briefly introduce the preliminaries required to follow this study before providing theoretical and experimental results. Mathematical proofs, simulation details, and algorithms can be found in Section A. Code will be made publicly available on GitHub for publication.

## 2 METHODS

### 2.1 PRELIMINARIES: SPIKING NEURON MODEL AND CAUSAL SUBNETWORKS

In this work, we focus on a special case of the widely used Leaky Integrate-and-Fire (LIF) neuron model: the IF model with exponential synapses (see Section A.1.1), also called the non-Leaky IF model (nLIF) (Mostafa, 2017; Göltz et al., 2021). A network of nLIF neurons is defined as follows:

**Definition 1 (nLIF)** *Let $L \in \mathbb{N}$, $\ell \in [1, L]$, $N_\ell \in \mathbb{N}$ be the number of neurons per layer $\ell$, $\tau_s \in \mathbb{R}^+$ be the synaptic time constant, $\vartheta \in \mathbb{R}$ be the threshold, and $t^{(0)} \in \mathbb{R}^{N_0}$, $N_0 \in \mathbb{N}$, be the inputs to the neural network. For $i \in [1, N_\ell]$, $j \in [1, N_{\ell-1}]$, let $W_{ij}^{(\ell)} \in \mathbb{R}$ be the synaptic weights from layer $\ell - 1$ to $\ell$. Then the membrane potential $u_i^{(\ell)} \in \mathbb{R}$ of a neuron $i$ in layer $\ell$ at time $t \in \mathbb{R}$ is given by:*

$$u_i^{(\ell)}(t) = \sum_{t_j^{(\ell-1)} \leq t} W_{ij}^{(\ell)} \left[ 1 - \exp\left( -\frac{t - t_j^{(\ell-1)}}{\tau_s} \right) \right] . \tag{1}$$

*The spike time $t_i^{(\ell)}$ of a neuron $i$ in layer $\ell$ is defined as $t_i^{(\ell)} = \inf\{t : u_i^{(\ell)}(t) = \vartheta\}$.*

Furthermore, we assume a widely used purely time-dependent encoding scheme in which each neuron spikes at most once (Comsa et al., 2020; Göltz et al., 2021; Stanojevic et al., 2023; Göltz et al., 2025; Che et al., 2024; Klos & Memmesheimer, 2025). This setup is motivated by two key considerations: **(i)** The nLIF neuron model is analytically tractable, enabling exact theoretical results. **(ii)** It is closely related to the LIF neuron model, providing a clear conceptual basis for generalising the reported results in future work.

The spike time of an nLIF neuron can be calculated analytically by finding, given a set of input spike times and weights, the corresponding *causal set*. The causal set contains the indices of all pre-synaptic neurons that cause the output spike time, i.e., its the set of neurons whose input spikes occur before the output spike. All input neurons with spike times larger than the output spike time do not affect it, and are hence not in the causal set. Formally, we define:

**Definition 2 (Causal set)** *Let $t_i^{(\ell)} \in \mathbb{R} \cup \{\infty\}$ be the spike time of a neuron receiving $N_{\ell-1} \in \mathbb{N}$ input spikes at times $t_j^{(\ell-1)}$ for $j \in [1, N_{\ell-1}]$. Then the corresponding causal set is given by $\mathcal{C}_i^{(\ell)}(t_1^{(\ell-1)}, ..., t_{N_{\ell-1}}^{(\ell-1)}) = \{j : t_j^{(\ell-1)} \leq t_i^{(\ell)}\}$ if $t_i^{(\ell)} < \infty$ & $\mathcal{C}_i^{(\ell)}(t_1^{(\ell-1)}, ..., t_{N_{\ell-1}}^{(\ell-1)}) = \emptyset$ otherwise.*

Although the causal set is typically represented as an unordered set in the literature, we define it here as an ordered set based on the indices $j$. Moreover, it implicitly depends on $W_i^{(\ell)}$ through $t_i^{(\ell)}$. If we know the causal set $\mathcal{C}_i^{(\ell)}$, the corresponding output spike time $t_i^{(\ell)}$ is given by (Mostafa, 2017)

$$t_i^{(\ell)} = \begin{cases} \tau_s \ln\left( \sum_{j \in \mathcal{C}_i^{(\ell)}} W_{ij}^{(\ell)} e^{t_j^{(\ell-1)}/\tau_s} \right) - \tau_s \ln\left( \sum_{j \in \mathcal{C}_i^{(\ell)}} W_{ij}^{(\ell)} - \vartheta \right) & \text{if } \mathcal{C}_i^{(\ell)} \neq \emptyset, \\ \infty, & \text{else}, \end{cases} \tag{2}$$

where the spike time is set to infinity if the input does not cause the neuron to spike. To find the causal set, we use the iterative approach described in Section A.1.2.

For deep feedforward SNNs, the concept of causal sets is generalised as follows:

**Definition 3 (Causal subnetwork)** *Let $L \in \mathbb{N}$, $\ell \in [1, L]$, $N_\ell \in \mathbb{N}$, $N_0 \in \mathbb{N}$. Further, let $\mathcal{C}_i^{(m)}$ be the causal set of neuron $i \in [1, N_m]$ in layer $m \in [1, L]$. Then for a subset $I \subseteq [1, N_\ell]$ of neurons in layer $\ell$, the causal subnetwork $\mathcal{P}_I^{(\ell)}(t^{(0)})$ given inputs $t^{(0)} \in \mathbb{R}^{N_0}$ is defined recursively:*

$$\mathcal{P}_{I,n-1}^{(\ell)} = \left( \mathcal{C}_j^{(n-1)} : j \in \mathcal{C} \text{ for } \mathcal{C} \in \mathcal{P}_{i,n}^{(\ell)} \right) \quad \text{with} \quad \mathcal{P}_{I,\ell}^{(\ell)} = \left( \mathcal{C}_i^{(\ell)} : i \in I \right) \quad \text{and} \quad n \in [1, \ell] . \tag{3}$$

As depicted in Fig. 1A, a causal subnetwork refers to the subset of neurons and connections that influenced the output spike times of neurons $i \in I$ of layer $\ell$, given inputs $t^{(0)}$. In this work, we represent it as a list of lists: for each layer, we include a list containing the causal sets of neurons in this layer that contributed to the spike times of neurons in $I$. It can be calculated from the observed spike times and connectome alone (algorithm in Section A.4.9), but depends implicitly on the weights.

## 2.2 CAUSAL PIECES: DEFINITION AND PROPERTIES

We introduce the concept of causal pieces, which we later demonstrate to be a useful tool for analysing the computational properties of SNNs. For a subset $I$ of neurons in layer $\ell$ of a feedforward SNN, the causal piece is a region in the joint input and parameter space for which the causal subnetwork remains fixed, meaning that the spike times of neurons $i \in I$ depend on the same subnetwork (Fig. 1A, bottom) within this region. Formally, using Definitions 1 to 3 we define a causal piece as follows:

**Definition 4 (Causal piece)** *Let $L \in \mathbb{N}$, $\ell \in [1, L]$, $N_\ell \in \mathbb{N}$, $t_0 \in \mathbb{R}^{N_0}$ be the input spike times to the network with $N_0 \in \mathbb{N}$, and $W \in \mathbb{W} = \mathbb{R}^{N_0 \cdot N_1} \times \ldots \times \mathbb{R}^{N_{L-1} \cdot N_L}$ the weights. Then for a subset $I \subseteq [1, N_\ell]$ of neurons from layer $\ell \in [1, L]$, we call $\mathbb{P}[\mathcal{P}_I^{(\ell)}]$ the causal piece associated to $\mathcal{P}_I^{(\ell)}$:*

$$\mathbb{P}[\mathcal{P}_I^{(\ell)}] = \{(t_0, W) \in \mathbb{R}^{N_0} \times \mathbb{W}: \text{ given } t_0 \text{ \& } W, \text{ the neurons } i \in I \text{ have causal subnetwork } \mathcal{P}_I^{(\ell)}\}$$

Throughout this paper, we often consider causal pieces for networks with fixed weights. In such cases, the causal piece is only defined by the inputs and reduces to $\mathbb{P}[\mathcal{P}_I^{(\ell)}] \subseteq \mathbb{R}^{N_0}$.

Under the perspective of Definition 4, the output spike times of an SNN are piecewise continuous, non-linear functions (Fig. 1B) of the inputs, with each region corresponding to a distinct causal piece. For nLIF networks, within each piece, the output spike times are Lipschitz continuous with respect to both the input spike times and weights – a useful property for gradient-based training. When transitioning between causal pieces – for example, by varying the input to an nLIF neuron – the output spike time may change continuously, discontinuously, or become undefined (i.e., the neuron ceases to spike), depending on how the causal sets of neurons change (see Section A.2.1 for details).

## 3 RESULTS

In the following, we first prove that the number of causal pieces provides a lower bound for the approximation error of an nLIF SNN. We then continue by demonstrating how to count them. The theoretical results are complemented by simulations, showing, in particular, that a high number of causal pieces on the training samples at initialisation correlates with training success (Fig. 3). Hence, the number of causal pieces can be used as an objective for optimising SNN initialisation in practice.

### 3.1 THE NUMBER OF CAUSAL PIECES IS A MEASURE OF EXPRESSIVITY

The approximation error of an nLIF network, i.e., how well a given function can be approximated, is lower bounded by an expression depending inversely on the number of causal pieces – meaning that more causal pieces result in potentially more expressive SNNs (for a proof, see Section A.3.3):

**Theorem 1 (Approximation bound)** *Let $-\infty < a < b < \infty$, $g \in C^3([a, b])$ so that $g$ is not affine. Then there exists a constant $c > 0$ that only depends on $\tau_s \int_a^b \sqrt{|\frac{\mathrm{d}^2}{\mathrm{d}x^2} e^{g(x)/\tau_s}|} \mathrm{d}x$ and a constant $\zeta > 0$ only depending on the maximum of $\max_x \left( e^{\Phi(x)/\tau_s} \right)$ and $\max_x \left( e^{g(x)/\tau_s} \right)$ so that*

$$\|\Phi - g\|_{L^\infty([a,b])} > \frac{c}{\zeta} p^{-2} \tag{4}$$

*for all nLIF neural networks $\Phi$ with $p$ number of causal pieces and time constant $\tau_s$.*

The theorem provides a local measure of expressivity for SNNs, valid for high-dimensional inputs along any line. Moreover, it is valid even if the output of the nLIF network $\Phi$ has any discontinuous behaviour. Although the proof is for single-spike neurons, it can be extended to nLIF neurons that spike multiple times and have a simple reset mechanism (see Section A.3.4). Still, for clarity and tractability, we focus on the single-spike case in this work. It also has to be noted that having many pieces does not translate into the SNN generalising well, for which fewer pieces might be favourable.

### 3.2 ESTIMATING THE NUMBER OF CAUSAL PIECES

Since the number of causal pieces is a measure of the expressivity of nLIF neural networks, it is of substantial interest to estimate this number. As every causal piece is characterised by a unique causal

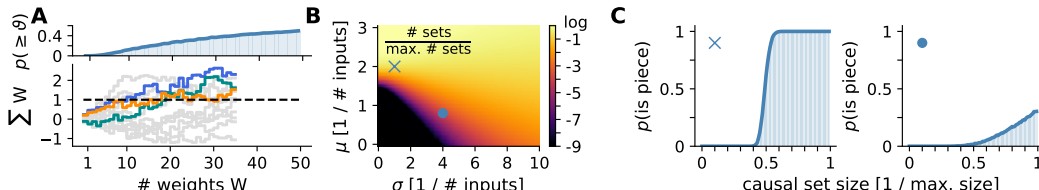

Figure 2: Estimating the number of causal pieces. **(A)** The probabilities $p_k^q$ are obtained by counting how many trajectories (cumulative sum of weights) are above the threshold at step $k$. The top panel shows the corresponding values of $p_k^q$, where $k$ is the number of weights. **(B)** Estimated number of pieces for weights sampled from normal distribution with different mean (y-axis) and standard deviation (x-axis). Colours are shown in log-scale. **(C)** $p_k^q$ for two points in (B), denoted by markers.

subnetwork, one way is to calculate the total number of causal subnetworks that can be formed. For a single nLIF neuron with $N$ total inputs, a naive upper bound for the number of causal pieces is therefore $2^N - 1$, which is the number of subsets that can be formed from a set of $N$ elements (minus the empty set).

However, not all of these subsets will be valid causal sets, e.g., the sum of the respective weights might not exceed the threshold. We obtain an improved upper bound by calculating the probability that, given weights sampled from a static random distribution $q$, the sum of $k$ weights exceeds the threshold, denoted by $p_k^q$. This is equivalent to the probability of a discrete random walk with continuous random step sizes (i.e. the weights) being above the threshold at step $k$ (Fig. 2A). This criterion is sufficient, as we can freely choose the inputs: all inputs of neurons in the causal set spike at the same time, while neurons not part of the set spike after the output neuron (Section A.3.5). The number of causal pieces $\eta^q$ is then upper bounded by:

$$\eta^q = \sum_{k=1}^{N} \binom{N}{k} p_k^q . \tag{5}$$

We show the improved upper bound of the number of sets as a fraction of $2^N - 1$ in Fig. 2B for weights randomly initialised from Gaussian distributions with different mean and variance (using a Monte Carlo approach, see Algorithm 1 in the appendix). For illustration purposes, $p_k^q$ is shown for two different $q$ in Fig. 2C. The obtained results highlight two points: **(i)** the highest number of causal pieces is reached only for distributions with non-zero mean – which is quite remarkable given that initialisation schemes in the literature, often borrowed from traditional deep learning, sample the weights from distributions with zero mean (Rossbroich et al., 2022; Bellec et al., 2018; Zenke & Vogels, 2021; Lee et al., 2016; Ding et al., 2022; Che et al., 2024). However, for extreme distributions, e.g., with very high mean, the estimated number of pieces is only achieved for data distributions that are very different from those found in practice, which will be discussed more thoroughly in the next section. **(ii)** With increasing variance, results tend to improve even if the mean is set non-optimally. In fact, one can show that in the limit of large variance, the number of pieces is lower bounded by an expression proportional to $N^{-3/2}$ (Theorem 2):

**Theorem 2 (Number of pieces in limit)** *Let $q$ be a symmetric probability distribution with mean $\mu < \infty$ and variance $\sigma^2$, and $W_j \sim q$ for $0 \leq j < N$. In the limit $\frac{\mu}{\sigma} \to 0$ and $\frac{\vartheta}{\sigma} \to 0$, the number of causal pieces is lower bounded by*

$$\eta^q \geq \frac{2^N - 1}{2N\sqrt{\pi \cdot (N - \frac{2}{3})}} , \tag{6}$$

which is, quite remarkably, valid for all probability distributions. This is a direct consequence of the Sparre Andersen theorem for random walks (Andersen, 1954; Majumdar, 2010), see Section A.3.6.

In case of deep SNNs, the number of causal pieces is equivalent to the number of routes on which spikes can flow unhindered from the inputs to the outputs through the network (Definition 3). For nLIF networks with $\{N_1, ..., N_\ell, 1\}$ neurons per layer, we find in Section A.3.7 that a naive upper bound for

the number of pieces of the output neuron is $\eta^q \leq 2^{\prod_{i=1}^{\ell} N_i} \leq 2^{N^{\ell}}$, where $N = \max\{N_1, ..., N_{\ell}, 1\}$. This is quite different from ReLU neural networks, which have an upper bound that scales only exponentially with the number of layers (Montufar et al., 2014) (or the total number of neurons (Hanin & Rolnick, 2019)). However, it remains to be seen whether networks with such a large number of pieces can be constructed, although Fig. 1C suggests quite dramatic increases in the number of causal pieces by adding even a single hidden layer.

### 3.3 THE PRACTICALLY RELEVANT NUMBER OF CAUSAL PIECES

In practice, even for single neurons we expect the number pieces to be below the improved bound we found, as most of these pieces will not be traversed when given realistic input data (i.e., not all inputs being identical). Moreover, the total number of pieces may be irrelevant for the learning problem at hand if a large fraction of the pieces occupy parts of the domain that are not populated by data. For example, Fig. 1C shows that the density of pieces can change dramatically throughout the domain. Thus, we propose an alternative approach to counting causal pieces which is more aligned to practical scenarios and less resource demanding: given a dataset, we count only the number of pieces that contain at least one data point. In the following, we demonstrate this for the Yin-Yang dataset (Kriener et al., 2022) using the standard scenario of 5000 random training samples, as well as by using a grid of inputs covering the whole input domain of the dataset (with 124980 samples in total). Yin-Yang is an ideal dataset for probing smaller neural networks, as it combines simplicity with a learning task that clearly separates linear and non-linear models. It also allows us to visualise causal pieces over the whole data domain, which is unfeasible for high-dimensional data. In the following, we only use this approach to count the number of causal pieces. An algorithm for counting causal pieces is provided in Section A.4.9.

### 3.4 THE NUMBER OF PIECES AT INITIALISATION CORRELATES WITH TRAINING SUCCESS

The initialisation scheme of parameters is crucial for training both ANNs and SNNs. Although for SNNs, schemes derived experimentally or adopted from ANNs have been successfully applied, a recent study highlighted the lack of a principled approach for identifying initialisation schemes that

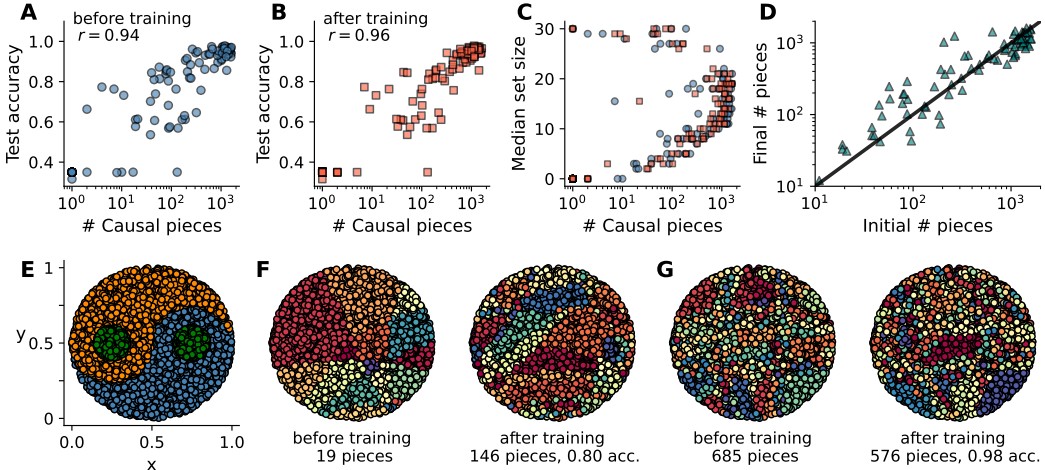

Figure 3: Network initialisation strongly affects training success. **(A)** The logarithm of the number of pieces (here: of the output layer) at network initialisation strongly correlates with performance after training ($r = 0.94$). The correlation between pieces and accuracy is $r = 0.77$. **(B)** Same as (A), but with the number of pieces after training. For pieces vs. accuracy, we find $r = 0.81$. **(C)** Median causal set size depending on the number of causal pieces before (blue) and after (red) training. **(D)** Number of pieces before and after training. The diagonal indicates no change in pieces. **(E)** Illustration of the Yin-Yang dataset with three classes: the two halves and the dots. **(F)** Causal pieces (each piece is indicated by a different colour) of a single output neuron for a bad initialisation, evaluated using only training samples. **(G)** Same as (F), but for one of the best initialisations.

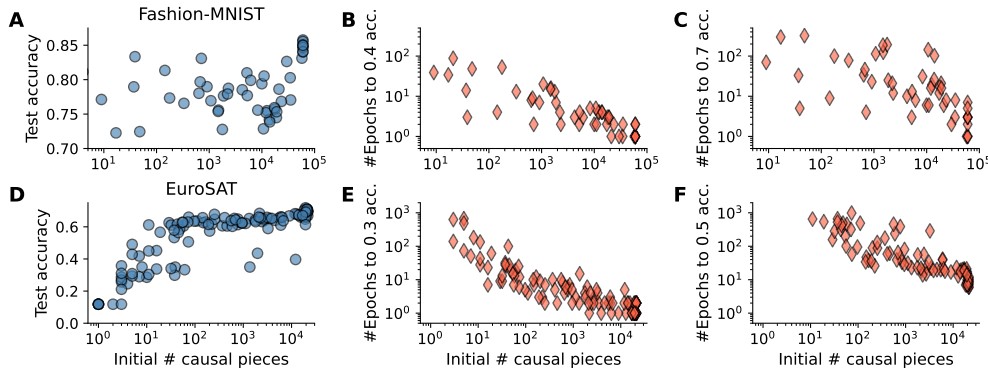

Figure 4: Experimental validation using Fashion-MNIST. **(A)** As in Fig. 3, the number of causal pieces the training data fall into at initialisation is correlated with the final test accuracy reached after training. For lower numbers of pieces, the final accuracy drops significantly. **(B)** Number of epochs required during training to reach a test accuracy of 40%, depending on the number of causal pieces the training data falls into at initialisation. **(C)** Same as B, but for a test accuracy of 70%. **(D-F)** Same as A-C, but for SNNs trained on the EuroSAT RGB dataset, and for 30% and 50% test accuracy. If a model did not reach the targeted accuracy level during training, no data points are shown in B-C and E-F.

facilitate the training of SNNs (Rossbroich et al., 2022). As a first application, we demonstrate that the number of causal pieces at initialisation, evaluated only using training samples, is a strong predictor of training success.

We trained 136 shallow nLIF networks with $[4, 30, 3]$ neurons using exact error backpropagation on analytically calculated spike times, as introduced in Mostafa (2017) (although we do not use any weight regularisation). To guarantee networks with a large variety of causal pieces after initialisation, we sampled weights from a normal distribution with randomly sampled mean and variance (see Section A.4). As shown in Fig. 3A,B, both the number of causal pieces of the last layer before and after training (evaluated using only training samples) strongly correlate with the final accuracy achieved on the test split. For networks with a high number of pieces, the causal pieces feature causal sets with a median size around $10 - 20$ elements (with 30 being the maximum), while networks with a low number of pieces have median set sizes that are either close to 0 or their maximum size. This is in agreement with Eq. (5), as the binomial coefficient has its maximum at $N/2$, while decreasing to 1 for $k = 0$ and $k = N$.

Interestingly, we find that it seems almost impossible to recover from a bad initialisation with low number of pieces through training (Fig. 3D). Networks with high number of causal pieces at initialisation will have a slightly reduced amount of pieces after training, while networks that start with a significantly lower number of pieces are not capable of reaching the number of pieces required for a high accuracy on the test set. Examples of the causal piece structure on the training data of the Yin-Yang dataset is shown for a single output neuron of a network achieving bad (Fig. 3F) and state-of-the-art performance (Fig. 3G) – clearly highlighting the difference in the number of causal pieces both before and after training.

A similar trend is observed for more complex datasets and larger networks as well, as shown in Fig. 4A for Fashion-MNIST. Here, we also see that the number of epochs required to reach a certain test accuracy is correlated with the number of causal pieces at initialisation, with SNNs that have many pieces learning much faster (Fig. 4B,C). In Fig. 4D-F, we show similar results for the EuroSAT RGB benchmark, a land cover classification task using Sentinel-2 satellite images.

Intuitively, a high number of pieces at initialisation means that the network is highly expressive and can more easily fit the training data. Moreover, it means that there are many different ways spikes can pass through the network, while a low number restricts the amount of routes – also making the collapse of pieces (i.e. no spiking) during training more severe. If an SNN starts with only a few pieces, the causal piece structure first has to be heavily restructured before proper learning can even happen, thus slowing down the training process. Hence, we argue that the number of causal pieces can be used as a measure for identifying good initialisation schemes for SNNs.

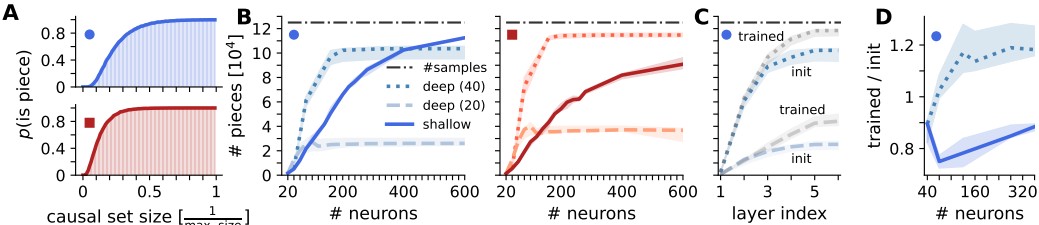

Figure 5: Width-and depth dependence of causal pieces. **(A)** $p_k^q$ of the optimised (top, dot) normal, and (bottom, square) uniform initialisation. **(B)** Number of pieces for shallow and deep networks. The maximum number, which is the number of input samples used to evaluate the number of causal pieces, is shown as a dash-dotted line. **(C)** Number of pieces per layer in a single network, before and after training. **(D)** Increase in the total number of pieces for deep and shallow networks. Markers denote results that belong together. We show medians (lines) and quartiles (shaded areas).

### 3.5 Increasing the number of pieces

As seen in the previous subsection, a large number of pieces is crucial to successfully train SNNs. Therefore, it is a natural question to ask through which means this number can be increased. From the previous results, an obvious option is to optimise the weight initialisation to yield networks with many pieces. We investigate this for networks ($[4, 100, 3]$ neurons) with weights initialised randomly from either a Gaussian or a uniform distribution, using a Yin-Yang dataset obtained from a $400 \times 400$ grid on the data domain. We chose a larger dataset here to properly probe the number of causal pieces. In case of a Gaussian distribution, the weights projecting into layer $\ell \in \mathbb{N}$, $W^{(\ell)} \in \mathbb{R}^{n_\ell \times n_{\ell-1}}$, are initialised by sampling from $\mathcal{N}(\alpha_0 \cdot n_{\ell-1}^{-\alpha_1}, [\alpha_2 \cdot n_{\ell-1}^{-\alpha_3}]^2)$ where $n_\ell$ is the number of neurons in layer $\ell$. Similarly, in case of a uniform distribution, weights are sampled from $\mathcal{U}(-v_0 + v_1, v_0 + v_1)$ with $v_0 = \beta_0 \cdot n_{\ell-1}^{-\beta_1}$ and $v_1 = \beta_2 \cdot n_{\ell-1}^{-\beta_3}$. The parameters $\alpha_i$ and $\beta_i$ ($i \in [0, 4]$) are found using a simple evolutionary algorithm that maximises the number of causal pieces (Section A.4.1). For this specific setup, we found $\alpha_0 = 1.69$, $\alpha_1 = 0.79$, $\alpha_2 = 1.13$, $\alpha_3 = 0.49$ and $\beta_0 = 1.85$, $\beta_1 = 0.39$, $\beta_2 = 1.02$, $\beta_3 = 0.54$. The corresponding probabilities $p_k^q$ of these weight initialisations are shown in Fig. 5A. As for the single neuron case, the weight distributions feature non-zero means. We visualise the causal pieces for a single output neuron in Fig. 6.

Another option to adjust the number of pieces is to change the width and depth of the SNN, as shown in Fig. 5B,C. We present three scenarios: (line) a shallow network where the width is steadily increased by increments of 20 neurons, (dashed) a deep network, where in each increment an additional hidden layer with 20 neurons is added, and (dotted) the same as for dashed, but with 40 neurons per hidden layer. Results are shown for the two distributions found using evolutionary optimisation. For the shallow network, the number of pieces grows consistently with increased network width, although slower than for deep networks and with a saturation setting in for very wide networks. In case of deep networks, the number of pieces grows rapidly initially, but then stagnates to a constant number of causal pieces. The effect is more pronounced if the hidden layers are wider, with a much stronger increase and final number of causal pieces for the network with 40 neurons per layer. Different from the expected exponential increase, we rather see a logistic growth. In fact, fitting logistic curves of the form $\gamma_0/(\gamma_1 + e^{-\gamma_2 N})$ with $\gamma_i \in \mathbb{R}$ and $N$ the number of neurons, we get a median relative error of $4 \cdot 10^{-2}$ (shallow), $2 \cdot 10^{-2}$ (deep 20), and $2 \cdot 10^{-2}$ (deep 40) for the Gaussian initialisation, and $9 \cdot 10^{-2}$ (shallow), $2 \cdot 10^{-2}$ (deep 20), and $5 \cdot 10^{-3}$ (deep 40) for the uniform one. The saturation for (deep 20) might occur due to a diminishing effect of pieces being split by consecutive layers. For all other cases, saturation most likely occurs since we reach the maximum number of causal pieces that can be counted using the data samples.

In Fig. 5C, we show the number of pieces per layer for a network with 5 hidden layers. Similarly to how initially adding hidden layers increased the number of pieces drastically in Fig. 5B, the highest increase is seen in the first few layers, with diminishing returns in deeper layers. In contrast, if we compare the number of pieces per layer before and after training, we find a slight increase in the number of causal pieces for deep layers. If we just focus on the total number of pieces of the whole network, we find that shallow networks end up with less pieces than at initialisation, while deep

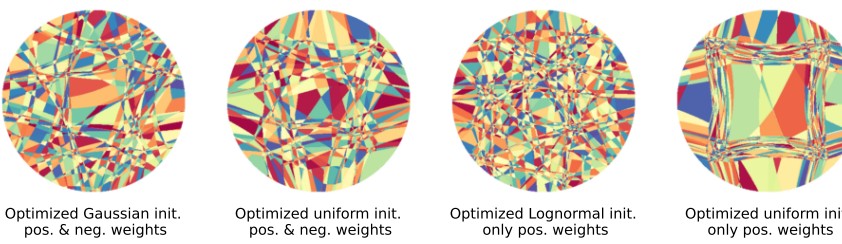

Optimized Gaussian init.
pos. & neg. weights

Optimized uniform init.
pos. & neg. weights

Optimized Lognormal init.
only pos. weights

Optimized uniform init.
only pos. weights

Figure 6: Causal pieces (coloured regions) of one of the output neurons for an nLIF neural network with $[4, 30, 3]$ neurons, using the initialisations obtained through evolutionary optimisation (Fig. 5 and Fig. 7). Causal pieces are evaluated using a $400 \times 400$ grid on the data domain.

networks end up with more (Fig. 5D). Most likely, this is because in a deep network, the number of pieces can be optimised by improving the misalignment of pieces between consecutive layers.

### 3.6 SPIKING NEURAL NETWORKS WITH EXCLUSIVELY POSITIVE WEIGHTS

Inspired by (Neuman et al., 2024), we study the case of SNNs with only excitatory neurons. In the mammalian neocortex, around 80% (Nieuwenhuys, 1994) of neurons are excitatory, i.e., their synapses only excite other neurons, which is equivalent to neurons having only positive outgoing weights in our nLIF neural networks. Although having only positive weights seems limiting at first, it comes with a significant advantage: controlling for continuity between linear pieces becomes much easier. In fact, the network is globally Lipschitz continuous as long as for each neuron, the input weights have a sum larger than the threshold – which can be easily enforced during training, e.g., through a regularisation term. The global Lipschitz constant of a neural network can be used to derive its covering number, which provides an upper bound for the network's generalisation error (Petersen & Zech, 2024). As seen from Theorem 3 in Section A.2.1, this bound can be improved by choosing network parameters that produce sparsely populated causal sets (small $|\mathcal{C}|$) that strongly overstep the threshold (large $\delta$). However, the contribution of the size of the causal sets in the Lipschitz constant is counter-balanced by the maximum weight $\bar{W}$, which has to be increased with decreasing set sizes to ensure that the sum of the weights exceeds the threshold.

We again optimise the parameters of two initialisation distributions, this time a lognormal and a uniform distribution – which both lead to networks with a similar number of pieces than for distributions with both postive and negative values. Their respective $p_k^q$ probabilities are shown in Fig. 7A. Using these initialisation schemes, we train networks composed of an SNN with positive weights and a single linear readout layer (with positive and negative weights, see Fig. 7C) on three different benchmarks: Yin-Yang, MNIST, and EuroSAT RGB – reaching in fact similar performance levels than other fully-connected ANNs, and far outcompeting linear models (Fig. 7D). An illustration of the causal pieces is shown in Fig. 6.

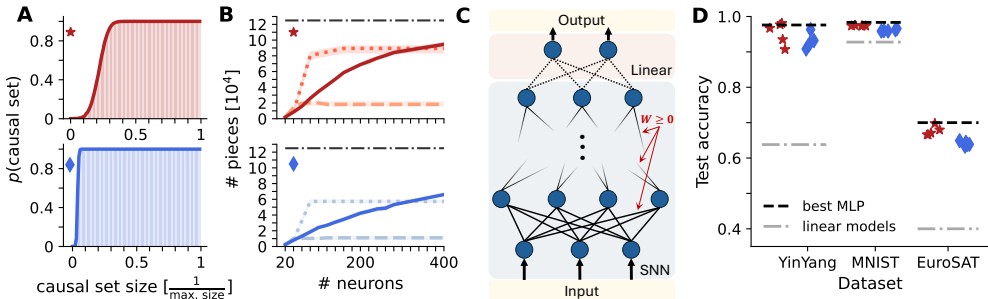

Figure 7: SNNs with only positive weights. **(A)** $p_k^q$ for (top, star) lognormal, and (bottom, diamond) uniform initialisation. **(B)** Number of pieces for shallow and deep networks. Labels as in Fig. 5B. **(C)** Used network architecture. **(D)** Performance on benchmarks. Each training run was repeated 5 times for different random seeds. Markers denote results that belong together.

## 4 DISCUSSION

We demonstrated that causal pieces are a promising tool for analysing and improving SNNs. One of our key findings is that the number of causal pieces at initialisation strongly correlates with SNN training success, making it a useful measure for identifying suitable initialisations. Across all reported experiments, we found that initialising weights from distributions with non-zero mean yields SNNs with the highest number of causal pieces – a strategy also used in Göltz et al. (2021) for single-spike LIF neurons. Moreover, the case of neural networks randomly initialised with only positive weights is very similar to having weights initialised from unconstrained distributions with non-zero mean, with both producing a comparable amount of pieces. In the introduced random walk picture, this is not too surprising, as both cases are drift-dominated random walks with (close to) 0 chance of returning to the threshold after passing it. Remarkably, this translates into SNNs with only positive weights (and a linear decoder) reaching comparable performance levels on standard benchmarks, although additional studies are required to properly analyse the benefits and limitations of such networks.

While previous work has briefly explored linear pieces in simplified spike-response models (Singh et al., 2023), our work is the first to lay the foundation for elevating this concept to more realistic neuron models. The decomposition of the input and parameter space into causal pieces should, in principle, generalise to any spiking neuron model commonly used in practice, especially since only the connectome and spikes of an SNN are required to obtain its causal pieces. Although our experiments focused on single-spike coding, we show that key results – such as the approximation bound in Theorem 1 – extend to the multi-spike setting. For spiking neurons with leak, the definition of causal sets, and hence causal subnetworks and pieces, remains unchanged (cf. Göltz et al. (2021); Comsa et al. (2020)). Therefore, causal pieces can be readily evaluated for LIF neurons with single-spike coding. However, mathematical results will require adapted proof strategies. For special choices of time constants, the output spike time of current-based LIF neurons can be calculated analytically using the Lambert W function Göltz et al. (2021), through which our proofs are likely to generalise. Similarly, while this work focused on feedforward architectures, we anticipate that results can be extended to recurrent SNNs by unrolling them in time, treating them as deep feedforward neural networks. Most importantly, the presented results directly apply to other simple neuron models as well, such as the simplified spike-response model used in Stanojevic et al. (2024). Generally, as long as the output spike time of a spiking neuron can be rewritten as a linear function of its inputs (e.g., through substitution of variables), the same proof strategy as presented in this work can be applied.

An important property of causal pieces, and neural networks in general, is their Lipschitz constant. The local Lipschitz constant of nLIF neural networks scales with the size of their causal sets, which is related to the number of synaptic interactions – a proxy measure for energy consumption in SNNs (Yin et al., 2021; Kucik & Meoni, 2021; Lunghi et al., 2024). Thus, the spike activity of SNNs might be directly tied to the learning task, i.e., the SNN requires more spikes for tasks with a high Lipschitz constant (and vice versa). Although we only briefly touched on Lipschitz constants in this work, we believe that this link might offer a novel data and model-dependent perspective on SNN design.

An even more important property of neural networks is their ability to generalise to previously unseen data. Causal pieces, like their counterparts *linear pieces* used for ANNs, are primarily a tool for assessing the approximation ability of neural networks though. Typical approaches for bounding the generalisation error of neural networks use covering numbers Neuman et al. (2024), which can only be calculated for neural networks that are globally Lipschitz continuous – a property that does not hold for SNNs in general. However, novel measures based on causal pieces, e.g., comparing the number of pieces the training and validation data fall into, might provide novel insights into the generalisation capabilities of SNNs.

To conclude, the presented results demonstrate that causal pieces are not only a powerful tool for increasing our understanding of SNNs, but also for guiding the design of improved network architectures and training methods. The causal piece framework naturally fits the discontinuous, event-based nature of SNNs. Most importantly, it enables a mathematically rigorous analysis of SNNs without requiring restrictive assumptions such as positive weights. We are confident that this approach will generalise to a wide range of neuron models and enable principled comparisons across spiking neuron models as well as with ReLU-based ANNs. Finally, we believe that the usefulness of causal pieces extends beyond technical applications and domains, potentially providing novel ways to study biological neurons by analysing their causal piece structure derived from experimental data.

ETHICS STATEMENT

All experiments were conducted using publicly available and widely accepted benchmark datasets. No personal or sensitive data were used, and there are no known ethical or bias concerns associated with the datasets.

REPRODUCIBILITY STATEMENT

Reproducibility of the results is ensured through several measures. All simulation experiments are described in detail in the appendix. A GitHub repository containing a Python package with the full implementation will be publicly released and cited in the final version. In addition, all experimental findings are supported by theoretical results, including formal mathematical proofs provided in the appendix.

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

# A APPENDIX

## A.1 METHODS

### A.1.1 RELATIONSHIP BETWEEN nLIF AND LIF NEURON MODELS

The current-based LIF neuron model with exponential synaptic kernel is given by

$$\frac{\mathrm{d}}{\mathrm{d}t} u_i^{(\ell)}(t) = \frac{1}{\tau_\mathrm{m}} (u_\mathrm{rest} - u_i^{(\ell)}(t)) + \frac{1}{\tau_\mathrm{s}} \sum_j W_{ij}^{(\ell)} \, \Theta \left( t - t_j^{(\ell-1)} \right) \exp \left( -\frac{t - t_j^{(\ell-1)}}{\tau_\mathrm{s}} \right) , \quad (7)$$

where $u_i^{(\ell)}(t) \in \mathbb{R}$ is the membrane potential of neuron $i$ in layer $\ell$ at time $t \in \mathbb{R}$, $W_{ij}^{(\ell)} \in \mathbb{R}$ is the synaptic weight connecting neuron $j$ of layer $\ell - 1$ to neuron $i$ of layer $\ell$, $t_j^{(\ell-1)}$ is the spike time of neuron $j$ in layer $\ell - 1$, $\tau_\mathrm{m} \in \mathbb{R}^+$ and $\tau_\mathrm{s} \in \mathbb{R}^+$ are the membrane and synaptic integration time constants, $\Theta \left( \cdot \right)$ is the Heaviside function, and $u_\mathrm{rest} \in \mathbb{R}$ is the rest value of the membrane potential.

In the special case $\tau_\mathrm{m} \gg \tau_\mathrm{s}$, this simplifies to

$$\frac{\mathrm{d}}{\mathrm{d}t} u_i^{(\ell)}(t) = \frac{1}{\tau_\mathrm{s}} \sum_j W_{ij}^{(\ell)} \, \Theta \left( t - t_j^{(\ell-1)} \right) \exp \left( -\frac{t - t_j^{(\ell-1)}}{\tau_\mathrm{s}} \right) , \quad (8)$$

which can be solved for $u_i^{(\ell)}(t)$ by integration:

$$u_i^{(\ell)}(t) = \int_{-\infty}^{t} \frac{\mathrm{d}}{\mathrm{d}t'} u_i^{(\ell)}(t') \, \mathrm{d}t' = \sum_{t_j^{(\ell-1)} \leq t} W_{ij}^{(\ell)} \left[ 1 - \exp \left( -\frac{t - t_j^{(\ell-1)}}{\tau_\mathrm{s}} \right) \right] . \quad (9)$$

### A.1.2 CALCULATING CAUSAL SETS

To find the causal set, we use the following approach: In case of an nLIF neuron that has $N_{\ell-1}$ input spike times $t_j^{(\ell-1)}$ with weights $W_{ij}^{(\ell)}$, we first define $\mathcal{K} = \{j_1, j_2, ..., j_{N_{\ell-1}}\}$ with $t_{j_1} \leq t_{j_2} \leq ... \leq t_{j_{N_{\ell-1}}}$. Furthermore, we set $\mathcal{K}_k = \{j_1, ..., j_k\}$ for $k > 0$. The causal set is then given by the subset $\mathcal{K}_m$ with the smallest index $m$ satisfying

**1.** $\sum_{j \in \mathcal{K}_m} W_{ij}^{(\ell)} \geq \vartheta$ **and 2.** $\mathcal{K}_m = \{j : t_j^{(\ell-1)} \leq t_i^{(\ell)}\}, t_i^{(\ell)} = \tau_\mathrm{s} \ln \left( \frac{\sum_{j \in \mathcal{K}_m} W_{ij}^{(\ell)} e^{t_j^{(\ell-1)} / \tau_\mathrm{s}}}{\sum_{j \in \mathcal{K}_m} W_{ij}^{(\ell)} - \vartheta} \right) .$

These two conditions are summarised as follows: (1) the inputs have to be strong enough to drive the membrane potential across the threshold, and (2) all inputs that did not cause the spike at time $t_i^{(\ell)}$ occur after it. The criterion of selecting the set with minimal $m$ ensures that we find the earliest possible output spike time. If no such set is found, the causal set is defined as the empty set, reflecting the fact that none of the inputs caused the neuron to spike. In simulations, we set the output spike time to a sufficiently large value such that it does not affect any other neuron in the network, emulating spiking at infinity.

## A.2 ADDITIONAL THEOREMS

### A.2.1 LIPSCHITZ CONTINUITY

To ease the notation, we drop the nested list notation of causal subnetworks in the following. We first state the result for a single nLIF neuron:

**Theorem 3 (Lipschitz continuous)** *Let $N_0 \in \mathbb{N}$, $j \in [1, N_0]$, and $\mathcal{C}_1^{(1)} \subset [1, \ldots, N_0]$. Moreover, let $a, b \in \mathbb{P}[\mathcal{C}_1^{(1)}]$ be the input to a single nLIF neuron with $N_0$ input times. Then the output spike time (Eq. (2)) is Lipschitz continuous with respect to input times and weights $W_{1j}^1 \in \mathbb{R}$, $j \in [1, N_0]$:*

$$\left\| t_1^{(1)}(a) - t_1^{(1)}(b) \right\|_{L^\infty(\mathbb{P}[\mathcal{C}_1^{(1)}])} \leq 2|\mathcal{C}_1^{(1)}| \max\left( \frac{\bar{W}}{\delta}, \frac{\tau_s}{\delta} \right) \|a - b\|_{L^\infty(\mathbb{P}[\mathcal{C}_1^{(1)}])} , \tag{10}$$

*where $|\mathcal{C}|$ denotes the cardinality of $\mathcal{C}$, $\|W_{1j}^{(1)}\| \leq \bar{W}$, $\delta < \sum_{j \in \mathcal{C}_1^{(1)}} W_{1j}^{(1)} - \vartheta$.*

The proof is given in Sections A.3.1 and A.3.2. In addition, the output spike time may change continuously, discontinuously, or become undefined when transitioning between causal pieces. Which of these occurs can be determined by inspecting the causal set: if all added or removed input neurons have identical spike times, the output spike time changes continuously; otherwise, it changes discontinuously. If the causal set would reach maximum size, but all inputs together do not reach the threshold, the output spike disappears. The corresponding result for entire networks follows from the fact that the composition of Lipschitz-continuous functions is itself Lipschitz continuous.

### A.3 MATHEMATICAL PROOFS

#### A.3.1 PROOF OF CONTINUITY AND DIFFERENTIABILITY

To improve readability, we drop the layer and output neuron indices in the following. First note that within a causal piece, the output spike time Eq. (2) is a composition of continuous and differentiable functions, and hence itself continuous and differentiable with respect to input spike times and weights.

In the following, we prove under which conditions the output spike time is a continuous function of input spike times and weights when crossing between neighbouring causal pieces. First, let $\mathcal{C}$ be the causal set of an nLIF neuron with input spike times $[t_0, \ldots, t_{N-1}]$, weights $[W_0, \ldots, W_{N-1}]$, and output spike time

$$t = \tau_s \ln\left(T\right) = \tau_s \ln\left( \frac{\sum_{j \in \mathcal{C}} W_j e^{t_j / \tau_s}}{\sum_{j \in \mathcal{C}} W_j - \vartheta} \right). \tag{11}$$

Let $\mathcal{C}'$ be the causal set of a neighbouring causal piece, with spike times $[\tilde{t}_0, \ldots, \tilde{t}_{N-1}, \tilde{t}_N]$, weights $[\tilde{W}_0, \ldots, \tilde{W}_{N-1}, \tilde{W}_N]$, and output spike time $\tilde{t}$:

$$\tilde{t} = \tau_s \ln\left(\tilde{T}\right) = \tau_s \ln\left( \frac{\sum_{j \in \mathcal{C}} \tilde{W}_j e^{\tilde{t}_j / \tau_s} + \tilde{W}_N e^{\tilde{t}_N / \tau_s}}{\sum_{j \in \mathcal{C}} \tilde{W}_j + \tilde{W}_N - \vartheta} \right). \tag{12}$$

We assume that the output spike time of $\mathcal{C}$ is along the border between the two causal pieces, meaning that $t = t_N$. Since output spike times can be shifted by $\Delta$ by shifting all input spike times by $\Delta$, without loss of generality, we assume that $\forall x \in \{t, \tilde{t}, t_0, \ldots, t_N, \tilde{t}_0, \ldots, \tilde{t}_N\}$, $x \geq 0$. All spike times are finite, thus $\exists t_{\max}$ with $0 < t_{\max} < \infty$ such that $\forall x \in \{t, \tilde{t}, t_0, \ldots, t_N, \tilde{t}_0, \ldots, \tilde{t}_N\}$, $x \leq t_{\max}$. Similarly, $\exists \bar{W} > 0$ such that $\forall \omega \in \{W_0, \ldots, W_N, \tilde{W}_0, \ldots, \tilde{W}_N\}$, $\|\omega\| \leq \bar{W}$. Furthermore, $\exists \epsilon_\vartheta$ with $0 < \epsilon_\vartheta < \infty$ such that $\epsilon_\vartheta < \sum_{j \in \mathcal{C}} \tilde{W}_j + \tilde{W}_N - \vartheta$. Lastly, we highlight the following identity:

$$T = T \cdot \frac{\sum_{j \in \mathcal{C}} W_j + M - \vartheta}{\sum_{j \in \mathcal{C}} W_j + M - \vartheta} \tag{13}$$

$$= T \cdot \frac{\sum_{j \in \mathcal{C}} W_j - \vartheta}{\sum_{j \in \mathcal{C}} W_j + M - \vartheta} + \frac{M \cdot T}{\sum_{j \in \mathcal{C}} W_j + M - \vartheta} \tag{14}$$

$$= \frac{\sum_{j \in \mathcal{C}} W_j e^{t_j / \tau_s}}{\sum_{j \in \mathcal{C}} W_j - \vartheta} \cdot \frac{\sum_{j \in \mathcal{C}} W_j - \vartheta}{\sum_{j \in \mathcal{C}} W_j + M - \vartheta} + \frac{M \cdot T}{\sum_{j \in \mathcal{C}} W_j + M - \vartheta} \tag{15}$$

$$= \frac{\sum_{j \in \mathcal{C}} W_j e^{t_j / \tau_s} + M \cdot e^{t_N / \tau_s}}{\sum_{j \in \mathcal{C}} W_j + M - \vartheta} \tag{16}$$

for all $M \in \mathbb{R}$ with $\sum_{j \in \mathcal{C}} W_j + M - \vartheta > 0$.

We first prove continuity for the argument of the logarithm by showing that $\forall \epsilon > 0, \exists \delta > 0$ such that $\|t_j - \tilde{t}_j\| < \delta$ with $j \in [0, N]$, $\|W_j - \tilde{W}_j\| < \delta$ with $j \in [0, N-1]$[1], and $\|T - \tilde{T}\| < \epsilon$. Using Eq. (16), we have:

$$\|T - \tilde{T}\| \tag{17}$$

$$= \left\| \frac{\sum_{j \in \mathcal{C}} W_j e^{t_j / \tau_s} + \sum_{j \in \mathcal{C}} \tilde{W}_j e^{t_N / \tau_s} + \tilde{W}_N e^{t_N / \tau_s} - \sum_{j \in \mathcal{C}} W_j e^{t_N / \tau_s}}{\sum_{j \in \mathcal{C}} \tilde{W}_j + \tilde{W}_N - \vartheta} \right.$$

$$\left. \frac{- \sum_{j \in \mathcal{C}} \tilde{W}_j e^{\tilde{t}_j / \tau_s} - \tilde{W}_N e^{\tilde{t}_N / \tau_s}}{\sum_{j \in \mathcal{C}} \tilde{W}_j + \tilde{W}_N - \vartheta} \right\| \tag{18}$$

$$\leq \frac{1}{\epsilon_\vartheta} \left( \|\tilde{W}_N\| \cdot \|e^{\tilde{t}_N / \tau_s} - e^{t_N / \tau_s}\| + \sum_{j \in \mathcal{C}} \|W_j\| \cdot \|e^{t_j / \tau_s} - e^{\tilde{t}_j / \tau_s}\| \right.$$

$$\left. + \|W_j - \tilde{W}_j\| \cdot \|e^{\tilde{t}_j / \tau_s} - e^{t_N / \tau_s}\| \right). \tag{19}$$

In the first step, we used Eq. (16) with $M = \sum_{j \in \mathcal{C}}(\tilde{W}_j - W_j) + \tilde{W}_N$, which leads to both $T$ and $\tilde{T}$ having the same denominator. Furthermore, we added the term $\sum_{j \in \mathcal{C}} W_j e^{\tilde{t}_j / \tau_s} - \sum_{j \in \mathcal{C}} W_j e^{\tilde{t}_j / \tau_s}$ in the numerator. In the next step, we used $\frac{1}{\epsilon_\vartheta} \geq \frac{1}{\sum_{j \in \mathcal{C}} \tilde{W}_j + \tilde{W}_N - \vartheta}$, and applied the triangle inequality several times. Using $\|\tilde{W}_j\| \leq \bar{W} \ \forall j \in [0, N]$, $\|e^{\tilde{t}_j / \tau_s} - e^{t_N / \tau_s}\| \leq \|1 - C\|$ with $C = e^{t_{\max} / \tau_s}$, and the mean value theorem for the exponential function, we then obtain:

$$\|T - \tilde{T}\| \leq \frac{C}{\epsilon_\vartheta \tau_s} \left( \sum_{j \in \mathcal{C}'} \bar{W} \|\tilde{t}_j - t_j\| + \sum_{j \in \mathcal{C}} \frac{\tau_s \|1 - C\|}{C} \|\tilde{W}_j - W_j\| \right). \tag{20}$$

Choosing $\|\tilde{W}_j - W_j\| < \delta_W$ with $\delta_W = \frac{\epsilon_\vartheta}{2N\|1-C\|} \cdot \epsilon$ and $\|\tilde{t}_j - t_j\| < \delta_t$ with $\delta_t = \frac{\epsilon_\vartheta \tau_s}{C \cdot \bar{W} \cdot 2(N+1)} \cdot \epsilon$, we arrive at

$$\|T - \tilde{T}\| < \epsilon. \tag{21}$$

The proof concludes by setting $\delta = \min(\delta_W, \delta_t)$. Continuity of the spike times then follows from the fact that the concatenation of continuous functions is again a continuous function.

Here we assumed that the neighbouring causal set $\mathcal{C}'$ has the property $\sum_{j \in \mathcal{C}'} \tilde{W}_j - \vartheta > 0$. If this is not the case, then at least one more input neuron with spike time $t^* = \min_x \{t_x \mid x \in \mathcal{K} \setminus \mathcal{C}'\}$ (with $t^* > t$) has to be added to the causal set until the condition holds again. Since the new output spike time has to be larger than $t^*$, its value jumps and is therefore not continuous when passing between causal pieces.

### A.3.2 LIPSCHITZ CONSTANTS

To improve readability, we drop the layer and output neuron indices in the following. Within a causal piece $\mathcal{C}$, the causal set does not change and the output spike time $t^*$ (Eq. (2)) is a composition of continuous and differentiable functions, and is therefore also continuous and differentiable. Hence, we estimate the Lipschitz constant by bounding the first derivative of the output spike time $t^*$.

Let $\mathcal{C}$ be a causal set with corresponding input spike times $t_0, ..., t_{N-1}$ for $N \in \mathbb{N}$, weights $W_0, ..., W_{N-1}$, and output spike time $t^*$. As in the previous subsection, we assume an upper bound for the absolute value of the weights, i.e., $\exists \bar{W} > 0$ such that $\forall \omega \in \{W_0, ..., W_{N-1}\}, \|x\| \leq \bar{W}$. Moreover, we assume that all spike times are larger or equal to 0, and we choose a $\delta > 0$ such that $\delta \leq \sum_j W_j - \vartheta$.

---

[1]Note that $W_N$ and $\tilde{W}_N$ cannot cause a switch between the two causal sets.

We first calculate the Lipschitz constant with respect to input spike times:

$$\left\| \frac{\partial t^*}{\partial t_k} \right\| = \left\| \frac{\partial}{\partial t_k} \tau_s \ln \left( \frac{\sum_{j \in \mathcal{C}} W_j e^{t_j / \tau_s}}{\sum_{j \in \mathcal{C}} W_j - \vartheta} \right) \right\| \tag{22}$$

$$= e^{-t^* / \tau_s} \left\| \frac{W_k e^{t_k / \tau_s}}{\sum_j W_j - \vartheta} \right\| \tag{23}$$

$$\leq \frac{\bar{W}}{\delta}, \tag{24}$$

where we used that $e^{(t_k - t^*) / \tau_s} \leq 1$ since $t^* \geq t_k$ by definition.

For weights, we get:

$$\left\| \frac{\partial t^*}{\partial W_k} \right\| = \left\| \frac{\partial}{\partial W_k} \tau_s \ln \left( \frac{\sum_{j \in \mathcal{C}} W_j e^{t_j / \tau_s}}{\sum_{j \in \mathcal{C}} W_j - \vartheta} \right) \right\| \tag{25}$$

$$= \tau_s e^{-t^* / \tau_s} \left\| \frac{e^{t_k / \tau_s}}{\sum_j W_j - \vartheta} - \frac{\sum_{j \in \mathcal{C}} W_j e^{t_j / \tau_s}}{(\sum_j W_j - \vartheta)^2} \right\| \tag{26}$$

$$= \tau_s e^{-t^* / \tau_s} \left\| \frac{e^{t_k / \tau_s} - e^{t^* / \tau_s}}{\sum_j W_j - \vartheta} \right\| \tag{27}$$

$$= \tau_s \left\| \frac{e^{(t_k - t^*) / \tau_s} - 1}{\sum_j W_j - \vartheta} \right\| \tag{28}$$

$$\leq \frac{\tau_s}{\delta}, \tag{29}$$

where we used that $0 \leq e^{(t_k - t^*) / \tau_s} \leq 1$ by definition, and hence $\| e^{(t_k - t^*) / \tau_s} - 1 \| \leq 1$.

Thus, for a causal piece $\mathbb{P}_{\mathcal{C}} \subseteq \mathbb{R}^{d \times d}$, where $d \in \mathbb{N}$ is the dimension of the input, and $a, b \in \mathbb{P}_{\mathcal{C}}$ we have:

$$\| t(a) - t(b) \|_{L^\infty(\mathbb{P}_{\mathcal{C}})} \leq 2 |\mathcal{C}| \max \left( \frac{\bar{W}}{\delta}, \frac{\tau_s}{\delta} \right) \| a - b \|_{L^\infty(\mathbb{P}_{\mathcal{C}})} \tag{30}$$

where $L_{\mathbb{P}_{\mathcal{C}}} = 2 |\mathcal{C}| \max \left( \frac{\bar{W}}{\delta}, \frac{\tau_s}{\delta} \right)$ is the Lipschitz constant of causal piece $\mathbb{P}_{\mathcal{C}}$ with causal set $\mathcal{C}$, and $|\mathcal{C}|$ is the number of elements in the causal set.

### A.3.3 PROOF OF THEOREM 1

To improve readability, we drop the layer indices in the following. First, we recapitulate the following theorem which holds, for example, for ReLU neural networks (Frenzen et al., 2010) (Theorem 2)[2]:

**Theorem 4** *Let* $-\infty < a < b < \infty$, $f \in C^3([a, b])$ *and* $f$ *is not affine. Then there exists a constant* $c > 0$ *that only depends on* $\int_a^b \sqrt{|f''(x)|} dx$ *so that*

$$\| \psi - f \|_{L^\infty([a,b])} > c \cdot p^{-2} \tag{31}$$

*for all piecewise linear* $\psi$ *with* $p \in \mathbb{N}$ *number of linear pieces.*

Eq. (2) can be written as a piecewise linear function by substituting $T_i = e^{t_i / \tau_s}$ (Mostafa, 2017), leading to:

$$T_i = \frac{1}{\sum_{j \in \mathcal{C}_i} W_{ij} - \vartheta} \cdot \sum_{k \in \mathcal{C}_i} W_{ik} T_k. \tag{32}$$

---

[2]See also Petersen & Zech (2024), Theorem 6.2

An nLIF neural network $\Psi(x)$ using this substitution is a composition of piecewise linear functions, and hence also itself a piecewise linear function. In this case, Theorem 4 applies to $\Psi$. The output of an equivalent nLIF network $\Phi$ without substitution is given by $\Phi = \tau_{\mathrm{s}} \ln \Psi$, i.e., we only apply the logarithm to the final output and scale by $\tau_{\mathrm{s}}$. This can be used to derive Theorem 1:

$$\|\Phi - g\|_{L^\infty([a,b])} = \tau_{\mathrm{s}} \left\| \ln \Psi - \ln \left( e^{g/\tau_{\mathrm{s}}} \right) \right\|_{L^\infty([a,b])}, \tag{33}$$

$$\geq \frac{\tau_{\mathrm{s}}}{\zeta} \| \Psi - e^{g/\tau_{\mathrm{s}}} \|_{L^\infty([a,b])}, \quad \text{with} \quad \zeta = \max \left[ \max_x(\Psi(x)), \max_x(e^{g(x)/\tau_{\mathrm{s}}}) \right], \tag{34}$$

$$> \frac{c}{\zeta} p^{-2}, \quad \text{with} \quad c > 0 \text{ depending only on } \tau_{\mathrm{s}} \int_a^b \sqrt{\left| \frac{\mathrm{d}^2}{\mathrm{d}x^2} e^{g(x)/\tau_{\mathrm{s}}} \right|} \mathrm{d}x, \tag{35}$$

where we applied the mean value theorem to arrive at Eq. (34) (i.e., we apply the mean value theorem to get rid of the logarithms) and Theorem 4 to arrive at Eq. (35). For the latter, we used the fact that if $g \in C^3([a,b])$ so that $g$ is not affine, then $e^{g/\tau_{\mathrm{s}}} \in C^3([a,b])$ is also not affine, allowing us to apply Theorem 4 using $f = e^{g/\tau_{\mathrm{s}}}$. Furthermore, we note that $\Phi$ and $\Psi$ have the same number of causal pieces.

### A.3.4 MULTIPLE-SPIKE CASE

We assume that after spiking, the membrane potential of the nLIF neuron is reset to its initial potential (here: $u_0 = 0$) and its dynamics continued. This way, the nLIF neuron can spike multiple times.

In this case, Theorem 1 can be generalised by showing that a multi-spike nLIF neuron can be represented by several single-spike nLIF neurons, one per output spike. Consequently, a network of multi-spike nLIF neurons can be mapped to a network of single-spike nLIF neurons, to which Theorem 1 can be applied. Thus, the theorem also applies to the multi-spike nLIF network.

We continue by showing how to represent a multi-spike nLIF neuron by several single-spike nLIF neurons. Assume a multi-spike nLIF neuron that spikes twice: at times $t_0$ and $t_1$. Moreover, let this neuron be part of a network, denoting its real-valued input weights by $W_{\mathrm{in}}$ and its output weights by $W_{\mathrm{out}}$. We can then replace this neuron by two single-spike nLIF neurons in the following way:

1. Add a single-spike nLIF neuron with initial potential $u_0 = 0$, connected to the same neurons as the multi-spike one via $W_{\mathrm{in}}$ and $W_{\mathrm{out}}$. This neuron fires at $t_0$, remaining silent thereafter.

2. Add another single-spike nLIF neuron with initial potential $u_0 = -\vartheta$. At time $t_0$, this neuron's membrane potential will be at 0, like the multi-spike neuron right after the reset. Consequently, the single-spike neuron will spike at time $t_1$.

If the multi-spike neuron spikes $n$ times, we can replace it by $n$ single-spike neurons with decreasing initial potentials $u_j(t = 0) = -j \cdot \vartheta$, $j \in [0, n-1]$.

### A.3.5 RANDOM WALKS

We drop the layer and output neuron index notation used in the main text to clear up the notation. Assume we have a single neuron with $N_0$ inputs. Let $\mathcal{K} = \{j_1, ..., j_K\} \subseteq [1, N_0]$ with $1 \leq K \leq N_0$, let $t_j$ be the input times and $W_j \in \mathbb{R}$ the corresponding weights, with $j \in [1, N_0]$. We denote by $p_k^q$ the probability that the subset $\mathcal{K}$ is a causal set if weights $W_j \sim q$ are sampled from a distribution $q$.

For $\mathcal{K}$ to be a causal set, we have to check the two conditions mentioned in Section 2. The first condition is satisfied if

$$\sum_{i \in \mathcal{K}} W_i \geq \vartheta. \tag{36}$$

Assuming the weights are sampled from a random distribution, this can be viewed as a random walk with discrete steps and randomly sampled, continuous step sizes. The position of the random walk at step $k$ is given by $S_k = \sum_{i=1}^k W_i$. In this framework, the first condition becomes the question of whether the random walk is above or equal to the threshold at step $K$, i.e., $S_K \geq \vartheta$.

The second condition – only spike times belonging to the causal set appearing before the output spike – can always be achieved by choosing inputs the following way (this does not apply to deep networks):

1. Set $t_j = c$ for $c \in \mathbb{R}$ and $j \in \{j_\ell, ..., j_K\}$.

2. Since condition 1 is satisfied, use Eq. (2) to calculate the output spike time $t$ with $\mathcal{K}$ as the causal set.

3. Set $t_j > t$ for $j \in \{j_\ell, ..., j_K\}$.

This way, any subset that suffices the first condition (sum of weights above threshold) is a valid causal set. Since we can choose inputs arbitrarily for a single nLIF neuron, $p_k^q$ is identical to the probability of the random walker to be above threshold at step $k$.

The values of $p_k^q$ are lower bounded by the first-passage-time distribution of the random walk. That's because the number of trajectories being above or equal to the threshold at step $k$ is lower-bounded by the number of trajectories that cross the threshold for the first time at step $k$.

### A.3.6 Proof of Theorem 2

Let $N \in \mathbb{N}$ be the number of inputs of a single nLIF neuron. We define $S_n = \sum_{i=1}^n W_i$ as the cumulative sum of weights $W_i \in \mathbb{R}$ with $S_0 = 0$ and $0 \leq n \leq N$. For the proof, we first note that $p_n^q \geq p_{\mathrm{FPT}}(n)$, where $p_{\mathrm{FPT}}(n) = p(S_n \geq \vartheta, S_{n-1} < \vartheta, S_{n-2} < \vartheta, ..., S_1 < \vartheta)$ is the first-passage-time distribution (at step $n$) for a random walk with discrete steps and random continuous step sizes ($W_j \sim q$), see Section A.3.5.

In the assumed limit, the survival probability, i.e., not passing the threshold until step $n + 1$, is given by the Sparre Andersen theorem (Andersen, 1954; Majumdar, 2010):

$$Q(n) = p(S_n < \vartheta, S_{n-1} < \vartheta, ..., S_1 < \vartheta) = \frac{1}{2^{2n}} \binom{2n}{n}. \tag{37}$$

The first-passage-time probability for step $n + 1$ is obtained by taking the difference of survival probabilities:

$$p_{\mathrm{FPT}}(n+1) = Q(n) - Q(n+1) \tag{38}$$

$$= \frac{1}{2^{2n}} \binom{2n}{n} - \frac{1}{2^{2n+2}} \binom{2n+2}{n+1} \tag{39}$$

$$= \frac{1}{2^{2n+1}} \binom{2n}{n} \left[ 2 - \frac{(2n+2)(2n+1)}{2(n+1)(n+1)} \right] \tag{40}$$

$$= \frac{1}{2^{2n+1}} \binom{2n}{n} \left[ 2 - \frac{(2n+1)}{(n+1)} \right] \tag{41}$$

$$= \frac{1}{2^{2n+1}} \binom{2n}{n} \frac{1}{n+1} \tag{42}$$

$$= \frac{C_n}{2^{2n+1}}, \tag{43}$$

with the Catalan number $C_n = \frac{1}{n+1} \binom{2n}{n}$. Using a lower bound for the Catalan number (Johnson), we get:

$$p_{n+1}^q \geq p_{\mathrm{FPT}}(n+1) \geq \frac{1}{2(n+1)\sqrt{\pi \cdot \left(n + \frac{1}{3}\right)}}. \tag{44}$$

This expression is monotonically decreasing, hence it reaches its minimum value at $n = N - 1$:

$$p_{n+1}^q \geq \frac{1}{2N\sqrt{\pi \cdot \left(N - \frac{2}{3}\right)}}. \tag{45}$$

Using this, we can estimate the number of causal pieces:

$$\eta^q = \sum_{k=1}^{N} \binom{N}{k} p_k^q \tag{46}$$

$$\geq \sum_{k=1}^{N} \binom{N}{k} p_{\text{FPT}}(k) \tag{47}$$

$$\geq \frac{1}{2N\sqrt{\pi \cdot \left(N - \frac{2}{3}\right)}} \cdot \sum_{k=1}^{N} \binom{N}{k} \tag{48}$$

$$= \frac{2^N - 1}{2N\sqrt{\pi \cdot \left(N - \frac{2}{3}\right)}} . \tag{49}$$

### A.3.7 NUMBER OF PIECES

For a single nLIF neuron, the number of pieces is obtained combinatorially: given $N$ inputs to the neuron, we can create $\binom{N}{k}$ different subsets with $k$ entries from these neurons. We denote by $p_k^q$ the probability that, if weights are sampled from a probability distribution $q$, a subset of $k$ inputs forms a causal set. The total number of causal pieces is then obtained by summing up the contributions of subsets of different length:

$$\eta = \sum_{k=1}^{N} \binom{N}{k} p_k^q . \tag{50}$$

The upper bound is obtained by using $p_k^q \leq 1$ for all $k$, and therefore $\eta \leq \sum_{k=1}^{N} \binom{N}{k} = 2^N - 1$.

For deep networks, we first look at a 2-layer network with $\{N_1, N_2, 1\}$ neurons, where $N_1$ is the number of inputs to the network. Starting with the output neuron, we can construct a single causal piece as follows: first, we sample a set of $r$ inputs. From the analysis for single nLIF neurons, we know that $\binom{N_2}{r} p_r^{q_2}$ such sets exist. Next, we have to estimate the number of pieces of the $r$ selected input neurons, which are all given by $\eta_1 = \sum_{k=1}^{N_1} \binom{N_1}{k} p_k^{q_1}$. However, the causal piece of the output neuron changes if any of its $r$ selected input neurons change their causal set. Thus, the number of pieces is given by $\binom{N_2}{r} p_r^{q_2} \eta_1^r$ – assuming the best case where the pieces of the output neuron are maximally split up by the input neurons. The total number is then given by:

$$\eta_2 = \sum_{r=1}^{N_2} \binom{N_2}{r} p_r^{q_2} \eta_1^r . \tag{51}$$

More generally, we have:

$$\eta_n = \sum_{r=1}^{N_n} \binom{N_n}{r} p_r^{q_n} \eta_{n-1}^r , \tag{52}$$

for $0 < n \leq \ell$ and $\eta_0 = 1$, where $\ell$ is the number of layers. Using $p_r^{q_n} \leq 1$ for all $n$ and $r$ and the binomial formula, we get:

$$\eta_n \leq \eta_{n-1}^{N_\ell} . \tag{53}$$

Applying this starting with $n = \ell$ until we arrive at $n = 1$, we get:

$$\eta_l \leq 2^{\prod_{i=1}^{\ell} N_i} \tag{54}$$

$$\leq 2^{N^\ell} , \tag{55}$$

with $N = \max\{N_1, N_2, ..., N_\ell, 1\}$.

### A.4 SIMULATION DETAILS

In all simulations, we use $\tau_s = 0.5$ and $\vartheta = 1$. To implement deep learning models, we used pyTorch (Paszke, 2019). Simulations were run on VSC-5 Vienna Scientific Cluster infrastructure, using A40 GPUs and AMD Zen3 CPUs. In general, individual simulations are rather short, lasting from seconds to minutes. Training larger networks on big datasets takes usually less than an hour.

### A.4.1 Optimising Initialisations

To find optimised initialisation schemes, we use a simple evolutionary method: Starting with a list with four different sets for the initial parameters, $P \in \mathbb{R}^{4 \times 4}$, we perturb each set by adding a random value sampled from a normal distribution $\mathcal{N}(0, 0.1^2)$. We then use all eight sets of parameters to initialise nLIF neural networks with weights sampled from our chosen distribution (e.g., normal, lognormal, uniform). For each network, we use the Yin Yang dataset (or any other method) to estimate the number of pieces. In this case, we sample the input space using a grid ($x \in [0, 1]$, $y \in [0, 1]$, 100 increments per dimension, constrained to the circular area). We then take the parameters that produced the four networks with the highest number of pieces and repeat this process, i.e., with using this new list as $P$. We stop if the number of pieces does not improve after $n \in \mathbb{N}$ loops.

For positive weights, we initialise weights using a lognormal distribution with mean $\alpha_0 \cdot n_{\ell-1}^{-\alpha_1}$ and standard deviation $\alpha_2 \cdot n_{\ell-1}^{-\alpha_3}$, or a uniform distribution $\mathcal{U}(v_0, v_0 + v_1)$ with $v_0 = \beta_0 \cdot n_{\ell-1}^{-\beta_1}$ and $v_1 = \beta_2 \cdot n_{\ell-1}^{-\beta_3}$. $n_{\ell-1}$ is the number of neuron projecting into layer $l$. Through the above optimisation loop, we found $\alpha_0 = 1.29$, $\alpha_1 = 0.57$, $\alpha_2 = 0.85$, $\alpha_3 = 0.76$ and $\beta_0 = 0.70$, $\beta_1 = 0.25$, $\beta_2 = 0.80$, $\beta_3 = 0.47$. The final parameters for normal and uniform (with positive and negative values) are provided in the main text.

### A.4.2 Details: Fig. 1

To initialise the networks, we use a normal distribution with the parameters found using evolutionary optimisation (see main text and Section A.4.1).

In panel B, the causal pieces of the output neuron of a network with $[10, 1]$ neurons is shown. For the plot shown top, we sample three random vectors $d_0 \sim \mathcal{N}(-2, 2^2)^{10}$, $d_1 \sim \mathcal{N}(-2, 2^2)^{10}$, $o \sim \mathcal{N}(-2, 2^2)^{10}$. The inputs $I$ are then obtained by spanning the plane using $I(\alpha_0, \alpha_1) = o + \alpha_0 \cdot (d_0 - o) + \alpha_1 \cdot (d_1 0 - o)$. We use $\alpha_0 \in [0, 1]$ and $\alpha_1 \in [0, 1]$ and 400 increments per variable. To get the line plot, we set $\alpha_1 = 0$ and increase $\alpha_0$ from 0 to 1 in 2000 increments.

In panel C, we use $d_0 \sim \mathcal{N}(0, 1)^{40}$, $d_1 \sim \mathcal{N}(0, 1)^{40}$, $o \sim \mathcal{N}(0, 1)^{40}$ and an increment of 400.

### A.4.3 Details: Fig. 2

To obtain the results, we used Algorithm 1 (see Section A.4.8) to estimate the number of pieces of a single nLIF neuron with weights sampled from $\mathcal{N}(\mu, \sigma^2)$. We ran the algorithm for values of $\mu$ and $\sigma$ ranging from 0 to 0.1 with increment 0.001. The maximum number of inputs was set to 100. For each initialisation, we sampled $10^4$ weight vectors (per $k$) to estimate $p_k^q$.

### A.4.4 Details: Fig. 3

For the normal distributions used to initialise the nLIF neural networks, the mean and standard deviation were both sampled from a uniform distribution $\mathcal{U}(-0.2, 0.8)$ and $\mathcal{U}(0, 1)$, respectively. Each reported data point corresponds to one sampled distribution. We calculate the number of causal pieces using only the 5000 training samples. We used the same grid to create the causal piece plots (panels F and G). Networks are trained using the Adam optimiser with a learning rate of $10^{-4}$ (no weight decay), batch size of 100, and 1000 epochs. The best test performance is reported.

As a loss function, we use the time-to-first-spike loss introduced in Göltz et al. (2021). For each sample $i$, its contribution to the loss is:

$$L_i = \log \left( \sum_{n=1}^{c} e^{(t_{i^*} - t_n)/\xi} \right), \tag{56}$$

where $c \in \mathbb{N}$ is the number of classes and $i^*$ is the correct label of sample $i$. $t_n$ is the output spike time of the output neuron encoding class $n$. We use $\xi = 0.2 \cdot \tau_s$. The final loss is obtained by averaging over all $N$ samples, $L = \frac{1}{N} \sum_{i=1}^{N} L_i$.

### A.4.5 Details: Fig. 4

For Fashion-MNIST, the trained networks have $[28 \times 28, 200, 100, 10]$ neurons, with the last layer being a standard linear readout layer. For the normal distributions used to initialise the nLIF

neural networks, the mean was sampled from a uniform distribution $\mathcal{U}(-\alpha, 1 - \alpha)$ with $\alpha \in [0.4, 0.5, 0.6, 0.7, 0.8]$ randomly selected. The standard derivation was sampled from $\mathcal{U}(0, 1)$. Each reported data point corresponds to one sampled distribution. We calculate the number of causal pieces using only the 60000 training samples. Networks are trained using the Adam optimiser with a learning rate of $10^{-3}$ (no weight decay), batch size of 100, and 400 epochs. As a loss function, we use the cross-entropy loss. The best test performance is reported.

The setup for EuroSAT RGB is the same, but we use a networks with $[16 \times 16 \times 3, 200, 100, 10]$ neurons. Moreover, the learning rate is set to $10^{-2}$ and the maximum number of epochs to 1000. The resolution of the EuroSAT images is decreased from $32 \times 32 \times 3$ to $16 \times 16 \times 3$.

### A.4.6 DETAILS: FIG. 5

For each data point, we show results of 10 runs with different random seeds. To calculate the number of causal pieces, we used an enlarged dataset composed of points obtained from a grid within the data domain, i.e., we evaluated the input space $[0, 1]^2$ using a $400 \times 400$ grid, leading to 124980 points (only points within the circular area were used). We obtained qualitatively similar results using a $600 \times 600$ grid. In panel C, we show the results for a network with $[4, 20, 20, 20, 20, 20, 3]$ and $[4, 40, 40, 40, 40, 40, 3]$ neurons (10 runs with different seeds). In panel D, the number of pieces of the output layer are shown for lognormal initialisation and (line) shallow networks with $[40, 80, 160, 320, 400]$ neurons in the hidden layer, as well as (dotted) deep networks with $[1, 2, 4, 5, 8, 10]$ hidden layers with 40 neurons each. Again the median over 10 runs with different random seeds is shown. For training, the same setup as described in Section A.4.4 was used.

### A.4.7 DETAILS: FIG. 7

Networks are initialised by sampling the weights either from a lognormal or uniform distribution, as described in Section A.4.1. To evaluate $p_k^q$, we again use the Monte Carlo approach described in Section A.4.8, with a similar setup as in Fig. 2. Panel B is created similarly as panel B in Fig. 5. To keep weights $W$ positive, we apply a ReLU function to them in the forward function, $W \mapsto \max(0, W)$.

For Yin Yang, we use a network of size $[4, 30, 3]$, with the last layer being a standard linear pyTorch layer. We train the networks using a batch size of 100, learning rate of $10^{-3}$, 5000 epochs, and Adam optimiser without weight decay. The reference values (0.638 and 0.976) are taken from Kriener et al. (2022) (best value also for $[4, 30, 3]$ neurons). They further report an accuracy of 0.855 if only the upper layer is trained, which is also lower than the performance reached by our networks.

For MNIST, we use a network of size $[28 \cdot 28, 200, 100, 10]$, again with the last layer being a standard linear pyTorch layer. Pixel values are re-scaled to be in the range $[0, 1]$. Images are flattened and no image transformations are used during training. We train the networks using a batch size of 100, learning rate of $10^{-3}$, 200 epochs, and Adam optimiser without weight decay. The best performance (0.9833) is taken from Kim et al. (2024). For the performance of a linear layer, we show 0.9277, as, e.g., reported in Senn et al. (2024).

For EuroSAT, we use a network of size $[16 \cdot 16, 200, 100, 10]$, again with the last layer being a standard linear pyTorch layer. Images are re-scaled to $16 \times 16$, with pixel values re-scaled to be in the range $[0, 1]$. Furthermore, we apply random horizontal and vertical flips during training. Images are flattened before they are provided as input to the neural networks. We train the networks using a batch size of 100, learning rate of $10^{-2}$, 1000 epochs, and Adam optimiser without weight decay. We found that the best performance of an MLP is similar to the one reached by random forests, which is 0.70. For the performance of linear models, we use the results achieved using logistic regression (0.40). We also reached 0.34 using nearest neighbor and 0.47 using decision trees.

### A.4.8 ALGORITHMS: MONTE CARLO APPROACH

In simulations, we use Algorithm 1 to calculate $p_k^q$, from which we calculate the improved upper bound using Eq. (5). A similar algorithm can be used to estimate $p_k^q$ for a static weight vector (with unknown distribution $q$) by randomly sampling subsets from the vector (e.g., in case of the weights in a trained neural network).

---

**Algorithm 1** Monte Carlo estimate for perceptron

---

**Require:** Distribution $q$, number of samples $num\_samples$, number of inputs $num\_inputs$, threshold $\vartheta$
1: $prob\_set \leftarrow$ list of length $num\_inputs$ filled with 0.      ▷ Probability that subset is a causal set.
2: **for** $causal\_set\_length = 1$ to $num\_inputs$ **do**
3:      **for** $sample\_ID = 1$ to $num\_samples$ **do**
4:          $W \leftarrow$ list of length $causal\_set\_length$ with values sampled from $q$
5:          $strong\_enough \leftarrow \sum_{i=0}^{num\_inputs-1} W_i \geq \vartheta$
6:          **if** $strong\_enough$ is $True$ **then**
7:              $prob\_set[causal\_set\_length] \leftarrow prob\_set[causal\_set\_length] + 1$
8:          **end if**
9:      **end for**
10:     $prob\_set[causal\_set\_length] \leftarrow prob\_set[causal\_set\_length] \,/\, num\_samples$
11: **end for**
12: **return** $prob\_set$

---

### A.4.9 ALGORITHMS: COUNTING PIECES

Algorithm 2 is used to count the number of causal pieces for (i) neurons in a deep neural network, and (ii) per layer. To count the pieces, we start from the first layer and index the causal sets. For neurons in the first layer, the causal sets are just composed of the inputs that caused the spike ((Algorithm 3, line 5). Each neuron's piece is given by the index we assign it (Algorithm 4). For neurons in deep layers, the causal set consists of both the indices of the inputs that caused it to spike, and the causal piece indices of these neurons (Algorithm 3, line 3). For layers (Algorithm 5), the causal set is given by the list of causal piece indices of all neurons in the layer. If any of these indices changes, the causal piece of the layer changes.

---

**Algorithm 2** Transform causal sets (per neuron) to causal piece IDs

---

**Require:** Nested list with ordered causal sets, `sets`. Dimensions are: samples, layers, neurons.
1: $causal\_set\_to\_ID \leftarrow$ empty dictionary
2: $causal\_set\_to\_ID[\text{String}([])] \leftarrow -1$
3: $num\_samples \leftarrow \text{length}(sets)$
4: $IDs \leftarrow$ list containing $num\_samples$ empty lists
5: **for** $sample\_id = 0$ to $num\_samples - 1$ **do**                  ▷ Iterate over samples
6:     $sets\_of\_sample \leftarrow sets[sample\_id]$
7:     **for** $layer\_id = 0$ to $\text{length}(sets\_of\_sample) - 1$ **do**          ▷ Iterate over layers
8:        $sets\_of\_layer \leftarrow sets\_of\_sample[layer\_id]$
9:        Append empty list to $IDs[sample\_id]$
10:       **for** each $causal\_set$ in $layers$ **do** ▷ Turn causal set of every neuron to corresponding ID
11:           $cset\_name \leftarrow \text{PROCESSCAUSALSET}(causal\_set, IDs, sample\_id, layer\_id)$
12:           $single\_ID \leftarrow \text{ASSIGNID}(cset\_name, causal\_set\_to\_ID)$
13:           Append $single\_ID$ to $IDs[sample\_id][layer\_id]$
14:       **end for**
15:     **end for**
16: **end for**
17: **return** $IDs$

---

**Algorithm 3** PROCESSCAUSALSET

---

**Require:** Causal set $causal\_set$, List of causal set IDs $IDs$, Sample index $sample\_id$, Layer index $layer\_id$
 1: **if** $layer\_id > 0$ **then**
 2:     $prev\_layer\_IDs \leftarrow IDs[sample\_id][layer\_id - 1]$
 3:     $cset\_name \leftarrow \text{String}([\text{Select from } prev\_layer\_IDs \text{ using } causal\_set, causal\_set])$
 4: **else**
 5:     $cset\_name \leftarrow \text{String}(causal\_set)$
 6: **end if**
 7: **if** $\text{length}(causal\_set) = 0$ **then**
 8:     $cset\_name \leftarrow \text{String}([])$
 9: **end if**
10: **return** $cset\_name$

---

**Algorithm 4** ASSIGNID

---

**Require:** Causal set name $cset\_name$, Dictionary $causal\_set\_to\_ID$
 1: **if** $cset\_name \notin \text{keys}(causal\_set\_to\_ID)$ **then**
 2:     $causal\_set\_to\_ID[cset\_name] \leftarrow \text{length}(causal\_set\_to\_ID)$
 3: **end if**
 4: **return** $causal\_set\_to\_ID[cset\_name]$

---

**Algorithm 5** Get Causal Piece ID for Neural Network Layers

---

**Require:** IDs, List of dictionaries `layer_indices_dict` with length $num\_layers - 1$
 1: $piece\_ID\_layers \leftarrow$ empty list
 2: **for** $sample\_ID = 0$ to $\text{length}(IDs) - 1$ **do**           ▷ Iterate over samples
 3:     Append empty list to $piece\_ID\_layers$
 4:     **for** $layer\_ID = 0$ to $\text{length}(IDs[sample\_ID]) - 1$ **do**     ▷ Iterate over layers
 5:         $lay\_state \leftarrow \text{String}(IDs[sample\_ID][layer\_ID])$
 6:         **if** $lay\_state \notin \text{keys}(layer\_indices\_dict[layer\_ID])$ **then**
 7:             $layer\_indices\_dict[layer\_ID][lay\_state] \leftarrow \text{length}(layer\_indices\_dict[layer\_ID])$
 8:         **end if**
 9:         Append $layer\_indices\_dict[layer\_ID][lay\_state]$ to $piece\_ID\_layers[sample\_ID]$
10:     **end for**
11: **end for**
12: **return** $piece\_ID\_layers$

---

