# OpenReview forum: "Causal pieces: analysing and improving spiking neural networks piece by piece"
_ICLR.cc/2026/Conference — Submitted to ICLR 2026_

### Official Review · Reviewer_tJr1 · 2025-10-31

**Soundness:** 2
**Presentation:** 2
**Contribution:** 1
**Rating:** 2
**Confidence:** 4

**Summary:**

This paper proposes Causal pieces that is a method to analyze the degree of expressivity of a given SNN. This method is theoretically backed only for a particular type of neurons (IF neurons with exponential synaptic inputs, which fire once at the most) though. It is found that the number of causal pieces is a good measure of SNN expressiveness and the learning capabilities.

**Strengths:**

Effort on theoretical understanding of SNNs. This paper theoretically attempts to understand SNNs (time-dependent models with rich dynamics), particularly, their expressiveness and learning capability, in terms of causal pieces. The deduction from theoretical understanding well aligns with the experimental results.

**Weaknesses:**

Considering simple models only: the proposed theoretical analysis is for IF neurons with exponential synaptic inputs, which fire merely once at the most. For realistic neurons that fire the unlimited number of spikes, it is not very straightforward to define causal subnets and pieces since they likely vary upon time.

Limited new findings out of theoretical understanding: The deductions from the understanding are not very surprising, like the relation between SNN expressiveness and number of causal pieces, learning capability and number of causal pieces, and the good learning performance for SNNs with positive weights only. The last one is quite clear even without this theoretical understanding since negative fan-in weights often cause dead neurons that degrade the expressiveness and learning capability. Further, the conclusion was drawn from SNNs on only toy datasets.

**Questions:**

Have the authors attempted to prove the theoretical deductions on datasets of high complexity?

---

> ### Author Response · Authors · 2025-11-19
>
> We would like to thank the reviewer for taking the time to read our manuscript and provide feedback on it. We have made several updates to the manuscript to address the mentioned concerns, in particular we added a new experiment for Fashion-MNIST (EDIT: we also added results for EuroSAT RGB). Below, we provide a point-by-point response to the comments and questions. We split up our response into two comments.
>
> **Weaknesses:**
>
> > Considering simple models only: the proposed theoretical analysis is for IF neurons with exponential synaptic inputs, which fire merely once at the most. For realistic neurons that fire the unlimited number of spikes, it is not very straightforward to define causal subnets and pieces since they likely vary upon time.
>
> First, as mentioned in the manuscript (below Theorem 2 and in the discussion), the derived results are also valid for nLIF with multiple-spiking. The proof is in the appendix, and is based on the fact that we can map any multi-spike network to a network of single-spike nLIF neurons.
>
> The core theoretical results are valid for the nLIF and related models. However, we would like to emphasize that this is true for most theoretical work on SNNs. Often, even simpler neuron models are used (such as the simple spike response model, or by constraining weights to be only positive).
>
> We chose the nLIF neuron model for two reasons:
> 1. The model can be treated analytically, which is a good starting point for deriving exact proofs.
> 2. It is a special case of the LIF neuron model, as for instance highlighted in [Goeltz et al., 2021]. Thus, it is conceptually closely linked to one of the most used spiking neuron models.
>
> Empirically, causal pieces can be calculated for leaky neurons as well. Basically: causal sets can be defined in a similar way (see [Goeltz et al., 2021]), from which the subnetworks, and hence causal pieces, can be calculated again. Moreover, in case of current-based LIF, for specific choices of the time constants (e.g., tau_syn = tau_mem), the output spike time can also be calculated analytically using the Lambert W function. Thus, an extension of our proof strategy is highly likely to work for more general cases.
>
> We added new parts in the Introduction and Discussion section to address these concerns.
>
> > Limited new findings out of theoretical understanding: The deductions from the understanding are not very surprising, like the relation between SNN expressiveness and number of causal pieces, learning capability and number of causal pieces, and the good learning performance for SNNs with positive weights only. The last one is quite clear even without this theoretical understanding since negative fan-in weights often cause dead neurons that degrade the expressiveness and learning capability. Further, the conclusion was drawn from SNNs on only toy datasets.
>
> We agree that some of the results are intuitive, which we actually see as a benefit, especially since it allows the concept to be easily applied to other spiking neuron models as well.
>
> Generally, most theoretical work on SNNs struggles with the discontinuous nature of spikes. This has been a problem for deriving learning rules, but also for deriving theoretical expressivity bounds. For instance, the output spike times of most threshold-based SNNs are not a continuous function of the input and network parameters, but can change discontinuously (or even disappear/reappear). The introduced causal piece framework naturally fits this discontinuous nature of SNNs and requires no simplifications to deal with it. We see this as a unique strength of our approach. We added a short paragraph in the Introduction to highlight this more.

---

> ### Author Response · Authors · 2025-11-19
>
> In addition, we would like to clarify a few points:
>
> 1. Although some results, such as that SNNs with more pieces are more expressive, seem intuitive or unsurprising, we here provide a first mathematical result where we show that this is actually the case. As far as we know, such a lower bound for SNNs has not been derived yet. Such work is also important, as intuition is a double-edged sword and can often be wrong or not universally true.
> 2. The high degree of intuition also stems from the fact that we put in a lot of effort into identifying a suitable definition of causal pieces such that these properties hold (and are potentially applicable to other neuron models as well).
> 3. We would like to emphasize that in all reported results, we show the test accuracy, not the training accuracy. We fixed the labels of the figures to avoid any confusion.
> 4. We show a clear correlation between the number of causal pieces at initialisation and the final obtained test accuracy; a result that is motivated by our derived theory, but clearly goes beyond it and has immediate practical relevance.
> 5. We further show that for 4), only the number of causal pieces the training data fall into are relevant. This simplifies the evaluation of causal pieces tremendously.
> 6. We also report on how the number of causal pieces changes per layer and before/after training. These results are quite different from what is expected from ReLU ANNs, where the number of pieces in randomly initialised networks scales the same for deep and shallow ANNs [Hanin & Rolnick, 2019].
>
> Regarding positive weights: we are not sure it is really that obvious, as positive-weight SNNs are rarely seen in the literature or used in practice. Even though discontinuities are easier to handle with only positive weights, the dynamical regime of the SNN is highly restricted, as neurons in the network can only excite each other - never inhibit. In fact, positive-weight only might increase the stability of training, but also potentially decrease its expressivity (the SNN basically becomes a purely monotonous function). That we obtain a similar number of causal pieces as for SNNs with positive and negative weights, when initialising such networks appropriately, is therefore remarkable. Also note that without proper initialisation, such SNNs learn incredibly slowly or not at all.
>
> **Questions:**
>
> > Have the authors attempted to prove the theoretical deductions on datasets of high complexity?
>
> Yes, we added simulations for the more complex Fashion-MNIST dataset that show a similar trend: a high number of pieces at initialisation leads to good training results, while lower counts lead to a drop in performance. We also show how the number of pieces at initialisation correlates with training speed. (EDIT: We further added results for the EuroSAT RGB dataset)

---

> > ### Author Response · Authors · 2025-11-21
> >
> > We made another update to the manuscript, adding new simulation results for EuroSAT RGB (Figure 4). The EuroSAT RGB benchmark is a land cover classification task using Sentinel-2 satellite images.

---

> > > ### Author Response · Authors · 2025-11-28
> > >
> > > Dear Reviewer,
> > >
> > > The end of the discussion period is drawing near, and we were wondering if you had any chance to think about our response.
> > >
> > > The initial review posed one question: "Have the authors attempted to prove the theoretical deductions on datasets of high complexity?" Indeed, this has happened now. We have added very comprehensive experiments that fully support our findings.
> > >
> > > In addition, the review critiques the lack of surprise in our findings. We have mentioned before that surprise is not a standard metric to assess research quality. It appears more appropriate to judge a result based on its correctness, its usefulness, or its rigorous presentation. I think it is clear at this point that these three points are satisfied, especially given the new large-scale experimental results.
> > >
> > > We would be delighted to hear your thoughts on this.

---

### Official Review · Reviewer_pLzs · 2025-10-31

**Soundness:** 3
**Presentation:** 3
**Contribution:** 2
**Rating:** 4
**Confidence:** 2

**Summary:**

The paper introduces “causal pieces,” a way to decompose spiking neural networks into locally Lipschitz regions, provides algorithms to count them, and proves that more pieces imply greater expressivity (under stated nLIF assumptions). Empirically, it shows that initial piece count strongly correlates with training success, that depth, especially early layers, inflates piece counts, and that positive weight SNNs can attain many pieces and competitive accuracy on simple benchmarks.

**Strengths:**

The paper provides up to five contributions.

The figures are beautiful, the theoretical proofs are solid, and the experiments align well with the claims.

**Weaknesses:**

Writing of Abstract and Introduction. The abstract looks messy. It mainly lists contributions and is almost the same as the second-to-last paragraph in the Introduction. I suggest improving the writing and readability of both sections.

“We believe that causal pieces are a powerful and principled tool for improving SNNs, and may also provide new ways of comparing SNNs and ANNs in the future.” The benefits of the proposed method for comparing SNNs and ANNs are not further described in the main text, which creates a mismatch between the abstract and the main paper.

**Questions:**

1. The writing of Section 3.4 should be reorganized. It describes the experimental setting and results after the demonstration and the argument derived from it.

2. Figure 3. The correlation study uses only a single dataset, which is insufficient to support the conclusion: “In particular, we demonstrate in simulation that parameter initialisations which yield a high number of causal pieces on the training set strongly correlate with SNN training success.”

3. Section 3.6. The results are credible for these experimental setups (flattened inputs, simple architectures), but “competitive” performance is shown only against specific baselines, not state-of-the-art spiking systems or modern vision-task backbones.

---

> ### Author Response · Authors · 2025-11-19
>
> First of all, we would like to extend our thanks to the reviewer for providing this honest and useful feedback on our manuscript. We tried to address all concerns and questions in our updated version. In particular, we added new simulation results on Fashion-MNIST (EDIT: we also added results for EuroSAT RGB). Below, we briefly answer all comments and questions.
>
> **Weaknesses:**
>
> > Writing of Abstract and Introduction. The abstract looks messy. It mainly lists contributions and is almost the same as the second-to-last paragraph in the Introduction. I suggest improving the writing and readability of both sections.
>
> Many thanks for this feedback. We have rewritten the abstract. It should now read much smoother. Of course, the main contributions of the paper are still listed, but we tried to provide more narrative glue between the individual points.
>
> We also updated the introduction to improve writing and readability. We are happy to further improve it given feedback.
>
> > “We believe that causal pieces are a powerful and principled tool for improving SNNs, and may also provide new ways of comparing SNNs and ANNs in the future.” The benefits of the proposed method for comparing SNNs and ANNs are not further described in the main text, which creates a mismatch between the abstract and the main paper.
>
> Many thanks, we agree that this sentence creates a mismatch between the abstract and the paper. We have removed the “future-looking” (“may also provide new ways of comparing SNNs and ANNs in the future”)  part of this sentence from the abstract.
>
> **Questions:**
>
> > The writing of Section 3.4 should be reorganized. It describes the experimental setting and results after the demonstration and the argument derived from it.
>
> Thanks for bringing this to our attention. We made a minor adjustment to the section. It now follows from Motivation -> Experimental setup and demonstration -> Explanation and take-away.
>
> > Figure 3. The correlation study uses only a single dataset, which is insufficient to support the conclusion: “In particular, we demonstrate in simulation that parameter initialisations which yield a high number of causal pieces on the training set strongly correlate with SNN training success.”
>
> To improve our argument, we added simulations  for the more complex Fashion-MNIST dataset that show a similar trend: a high number of pieces at initialisation leads to good training results, while lower counts lead to a drop in performance. We also show how the number of pieces at initialisation correlates with training speed. (EDIT: We further added results for the EuroSAT RGB dataset)
>
> > Section 3.6. The results are credible for these experimental setups (flattened inputs, simple architectures), but “competitive” performance is shown only against specific baselines, not state-of-the-art spiking systems or modern vision-task backbones.
>
> By competitive, we meant compared to the best results obtained by comparable architectures. In our case, this is fully-connected ReLU ANNs. As explained in the appendix, we looked up the best possible results for MLPs in the literature for all shown benchmark tasks. Please also note that we always report the test accuracy, not the training accuracy. We updated the figure labels to avoid any confusion.
>
> We do agree that the phrasing “competitive” can be misleading and is too imprecise, though. In the manuscript, we made our formulations more specific by stating compared to which type of models we reach “competitive” performance.

---

> > ### Author Response · Authors · 2025-11-21
> >
> > We made another update to the manuscript, adding new simulation results for EuroSAT RGB (Figure 4). The EuroSAT RGB benchmark is a land cover classification task using Sentinel-2 satellite images.

---

> > ### Comment · Reviewer_pLzs · 2025-11-24
> >
> > Thanks authors for the rebuttal and for providing new results. I will maintain the original score.

---

> > > ### Author Response · Authors · 2025-11-24
> > >
> > > Dear Reviewer,
> > >
> > > Thank you very much for your response.
> > >
> > > We sincerely aimed to address all of your suggestions, including revisions to the abstract and introduction, as well as adding two new experiments in response to your feedback. Given these substantial updates and your previous remark that *"the figures are beautiful, the theoretical proofs are solid, and the experiments align well with the claims"*, we were hoping for a reconsideration of the initial score, which currently sits below the acceptance threshold.
> > >
> > > If there are remaining concerns or specific areas where the submission still falls short, we would be very grateful if you could share them.
> > >
> > > Thank you again for your time and consideration.

---

> > > > ### Author Response · Authors · 2025-11-28
> > > >
> > > > Dear Reviewer,
> > > >
> > > > The end of the discussion period is drawing near, and we were wondering if you had any chance to think about our questions.
> > > >
> > > > As already mentioned in response to another reviewer, we believe that every reviewer question has been fully addressed, all weaknesses were resolved, and strong, large-scale new experiments were included.
> > > >
> > > > Again, we point out that there are new numerical experiments that you may not have seen, but that clearly show, on a large scale, the effectiveness of our results.
> > > >
> > > > We note that despite very positive initial reviews, you have not posed additional questions or modified your score in response to our updates. We would be really grateful if you could give a bit of insight as to why.

---

### Official Review · Reviewer_raxD · 2025-11-02

**Soundness:** 3
**Presentation:** 3
**Contribution:** 2
**Rating:** 4
**Confidence:** 3

**Summary:**

This paper introduces an innovative analytical tool called "Causal Pieces" for spiking neural networks (SNNs). The paper demonstrates that the number of causal pieces is a measure of the expressive power of an SNN, and experiments show that the number of causal pieces at network initialization is highly positively correlated with the final accuracy. Furthermore, the paper finds that even SNNs with only positive weights can exhibit a large number of causal pieces and achieve good performance on benchmark tasks.

**Strengths:**

- Introduces and formalizes the "causal piece," a novel and natural abstraction for SNNs that partitions the input-parameter space into discrete regions, each governed by an identical causal subnetwork.
- Provides rigorous proof that the number of causal pieces serves as a quantitative measure of an SNN’s expressive capacity, directly linking this number to the network's approximation bounds.
- Empirically demonstrates a strong positive correlation between the number of causal pieces at initialization and final accuracy, offering a principled and highly valuable metric for SNN initialization.

**Weaknesses:**

1. **Model-Specific Limitations**: All theoretical proofs and primary experiments are based on a simplified neuron model: the nLIF, which assumes no leakage and that each neuron fires at most once (single-spike coding).

2. **Questionable Generalizability**: While the authors suggest in the discussion that the "causal piece" concept could extend to more common and complex models (e.g., leaky LIF, multi-spike coding, and recurrent SNNs), the paper provides no rigorous proof or experimental support for these claims. This generalization currently remains speculative "future work."

3. **Expressivity vs. Generalization**: The paper proves a link between the number of causal pieces and approximation ability (fitting capability), correlating it with training accuracy. However, the authors rightly concede (end of Sec 3.1) that "having many pieces does not translate into the SNN generalising well." A network with an excessive number of pieces might simply be overfitting the training data. The relationship between causal piece count and the more critical metric of generalization remains unclear.

4. **Limited Experimental Validation**: The empirical evaluation is primarily restricted to relatively simple datasets (like MNIST and Yin Yang). It lacks a systematic comparison on larger-scale, temporally complex tasks against strong, established baselines.

5. **Finding is Conceptually Intuitive**: The positive correlation between the number of "causal pieces" (as a measure of partitioning complexity) and the network's expressive power is, at a high level, conceptually intuitive. While the paper provides a valuable, SNN-specific formalization, the core finding itself is not entirely surprising.

**Questions:**

1. **Computational complexity**: For modern large scale networks, what are the time and memory complexities of computing causal pieces with your method?

2. **Saturation vs exponential growth**: Why does the number of pieces in deep networks appear to saturate rather than grow exponentially?

3. **Training dynamics**: During training or across different samples, how does the total number of causal pieces change? Is there a sample level correlation between prediction accuracy and the assigned causal piece regions?

4. **Generalization on more complex datasets**: Beyond training accuracy, when controlling for model size and dataset size, what is the relationship between the number of pieces and test accuracy on datasets such as CIFAR100 or ImageNet?

---

> ### Author Response · Authors · 2025-11-19
>
> Many thanks for this well-structured and detailed feedback on our manuscript. In the following, we address all questions and comments point-by-point. Moreover, we updated the manuscript to address all concerns, in particular we added a new experiment for Fashion-MNIST (EDIT: we also added results for EuroSAT RGB). We also split up our reply in several comments.
>
> **Weaknesses:**
>
> > Model-Specific Limitations: All theoretical proofs and primary experiments are based on a simplified neuron model: the nLIF, which assumes no leakage and that each neuron fires at most once (single-spike coding).
>
> Yes, the core theoretical results are valid for the nLIF and related models. However, we would like to emphasize that this is true for most theoretical work on SNNs. Often, even simpler neuron models are used (such as the simple spike response model, or by constraining weights to be only positive).
>
> We chose the nLIF neuron model for two reasons:
> 1. The model can be treated analytically, which is a good starting point for deriving exact proofs.
> 2. It is a special case of the LIF neuron model, as for instance highlighted in [Goeltz et al., 2021]. Thus, it is conceptually closely linked to one of the most used spiking neuron models.
>
> As mentioned in the manuscript (below Theorem 2 and in the discussion), the derived results are also valid for nLIF with multiple-spiking. The proof is in the appendix, and is based on the fact that we can map any multi-spike network to a network of single-spike nLIF neurons.
>
> Regarding leak: indeed, the proof of Theorem 2 will need to be extended to account for this case, which is left for future work. However, empirically, causal pieces can be calculated for leaky neurons as well. Basically: causal sets can be defined in a similar way (see [Goeltz et al., 2021]), from which the subnetworks, and hence causal pieces, can be calculated again.
> Moreover, in case of current-based LIF, for specific choices of the time constants (e.g., tau_syn = tau_mem), the output spike time can also be calculated analytically using the Lambert W function. Thus, an extension of our proof strategy is highly likely to work for more general cases.
>
> Generally, most theoretical work on SNNs struggles with the discontinuous nature of spikes. This has been a problem for deriving learning rules, but also for deriving theoretical expressivity bounds. For instance, the output spike times of most threshold-based SNNs are not a continuous function of the input and network parameters, but can change discontinuously (or even disappear/reappear). The introduced causal piece framework naturally fits this discontinuous nature of SNNs and requires no simplifications to deal with it. We see this as a unique strength of our approach.
>
> We would also like to highlight that the results do hold for other neuron models as well, not just the nLIF model. For instance, the recently used spike response model with linearly growing kernel (as used in [Stanojevic et al. 2023 & 2024]) is a special neuron model for which our results hold. In principle, the proof of Theorem 2 can always be applied if, by substituting the spike times, each neuron can be written as a linear function of its inputs.
>
> We added comments reflecting the clarifications made above to the Introduction and Discussion section.
>
> > Questionable Generalizability: While the authors suggest in the discussion that the "causal piece" concept could extend to more common and complex models (e.g., leaky LIF, multi-spike coding, and recurrent SNNs), the paper provides no rigorous proof or experimental support for these claims. This generalization currently remains speculative "future work."
>
> Yes, the focus of this paper is to introduce the concept, ground it mathematically, and highlight its practical usefulness. Covering all varieties of SNN models is therefore out of scope for this paper, and usually also not done in other learning theory papers on SNNs. However, as mentioned in the previous comment, we purposefully chose the nLIF model to enable generalisation to LIF in future work.
>
> What we would like to highlight is that in order to calculate causal pieces, only the network connectome and the spikes of each neuron are needed. This is available for almost every spiking neuron model, and hence the way we define causal pieces is not tied to the neuron model chosen here. We added a brief part in the Discussion section to cover this.

---

> ### Author Response · Authors · 2025-11-19
>
> > Expressivity vs. Generalization: The paper proves a link between the number of causal pieces and approximation ability (fitting capability), correlating it with training accuracy. However, the authors rightly concede (end of Sec 3.1) that "having many pieces does not translate into the SNN generalising well." A network with an excessive number of pieces might simply be overfitting the training data. The relationship between causal piece count and the more critical metric of generalization remains unclear.
>
>
> First of all, we would like to point out that in all figures, we report the test accuracy (not the training accuracy). We fixed the labels in the manuscript to avoid any confusion. So our analysis does, in fact, look into how well the trained models perform on unseen data.
>
>
> Second, even for ReLU ANNs, the effect of linear pieces on generalisation is not yet understood. Similarly, we have so far no formal way of characterising the generalisation capabilities of SNNs using causal pieces. However, one might derive metrics based on how many pieces the training and validation data fall into (e.g., if both training and validation data fall into extremely many pieces, the model might be overly expressive!). Such new metrics are desperately needed to understand generalisation in data-sparse regimes.
>
> Furthermore, we would like to stress that bounding the generalisation error of SNNs even with established metrics is not trivial. A common approach for ANNs is via the covering number, which can only be calculated if the neural network is continuously parametrised. This is generally not the case for SNNs, and thus generalisation bounds have only been derived for neuron models even simpler than nLIF [Neuman et al., 2024].
>
> We added a brief paragraph to discuss this in the Discussion section.
>
> > Limited Experimental Validation: The empirical evaluation is primarily restricted to relatively simple datasets (like MNIST and Yin Yang). It lacks a systematic comparison on larger-scale, temporally complex tasks against strong, established baselines.
>
> Thanks for this feedback. To improve our argument, we added simulations for the more complex Fashion-MNIST dataset that show a similar trend: a high number of pieces at initialisation leads to good training results, while lower counts lead to a drop in performance. We also show how the number of pieces at initialisation correlates with training speed. (EDIT: We further added results for the EuroSAT RGB dataset).
>
> > Finding is Conceptually Intuitive: The positive correlation between the number of "causal pieces" (as a measure of partitioning complexity) and the network's expressive power is, at a high level, conceptually intuitive. While the paper provides a valuable, SNN-specific formalization, the core finding itself is not entirely surprising.
>
> We actually see this as a major strength: the conceptual simplicity allows it to be mapped to different spiking neuron models (including even experimental data), making this a broadly applicable tool.
>
> Moreover, we provide a proper lower bound of the approximation error that grounds this intuition in theory. As far as we are aware, such bounds / proofs have not yet been derived for SNNs. This is important, since often intuition can be deceiving.
>
> Lastly, while it is conceptually intuitive that more pieces lead to higher expressivity, many aspects remain quite unintuitive. For instance, how does one choose network parameters / architecture to maximise the number of pieces? How does the causal piece structure change during training? How can we improve learning algorithms to deal with bad causal piece structures? Etc.
>
> In this work, we tackle some of these questions, but also open them up to the community to investigate.
>
> **Questions:**
>
> > Computational complexity: For modern large scale networks, what are the time and memory complexities of computing causal pieces with your method?
>
> We will provide a brief model-agnostic estimate here. The complexity splits up into two parts:
>
> 1. We evaluate the number of causal pieces for the training data only. Thus, if we have N training samples, we have to perform N forward passes of our network.
> 2. To calculate the causal pieces, we need the causal set of every neuron. In our case, we already get this from the forward pass, as we solve the nLIF dynamics analytically. In any case, if the maximum width of the network is M, then each causal set can have at most M entries. Thus, we have at maximum L*M^2 entries (L = number of layers).
> 3. To get the causal subnetwork, we go, layer by layer, backwards through the network starting from the neurons in the last layer. For each neuron, we simply add its causal set to a list structure, and we do this in total M*L times.
>
> To save memory, we actually hash the causal subnetwork (a list of lists) we obtained for each data sample.

---

> ### Author Response · Authors · 2025-11-19
>
> > Saturation vs exponential growth: Why does the number of pieces in deep networks appear to saturate rather than grow exponentially?
>
> This is briefly addressed in the manuscript already. We provide a more detailed explanation here.
>
> There are two reasons:
>
> 1) Exponential growth is only the best possible case when stacking layers. For a large growth to appear, subsequent layers have to “split” the causal pieces of previous layers. Imagine we evaluate the causal pieces for a single layer of neurons (with respect to the input of the layer). In this case, we get a mosaic plot as shown in our manuscript, which changes if we change the input weights of the layer. If we stack two such layers, it is like “overlaying” two such mosaic images. If all pieces in the two layers overlap, the total number of causal pieces does not grow at all. If there is only partial overlap, the number of pieces grows (they get “split up”). But this depends on how the weights are chosen. So saturation occurs when pieces are no longer split further by subsequent layers.
>
> Generally, even for ANNs, often no exponential growth is seen in the number of linear pieces [Hanin & Rolnick, 2019].
>
> 2) We evaluated the number of pieces on the training data. Thus, we can only have (at max.) as many pieces as training samples, which naturally leads to saturation if our network is very expressive.
>
> > Training dynamics: During training or across different samples, how does the total number of causal pieces change? Is there a sample level correlation between prediction accuracy and the assigned causal piece regions?
>
> This is a very good question. This is partially covered in the paper, so we highlight our results here.
>
> 1. If the SNN starts with only few pieces, usually the number of pieces increases during training but never reaches a sufficiently high number (compared to networks that start very high and consequently reach good performance). We show this in Fig. 3 and 4.
> 2. If the SNN starts with many pieces, usually the number of pieces first goes up during training and then goes down again.
> 3. In Fig. 5C, we show how the number of pieces increases per layer before and after training an SNN. Before training, we see an increase in pieces per layer, which saturates for deep layers. After training, we see that the saturation was partially resolved: alignment of pieces of subsequent layers was improved, and thus the number of causal pieces in deeper layers increased.
> 4. In Fig. 5D, we show how the total number of causal pieces changes for deep and shallow network before and after training. For deep networks, the number of pieces is usually higher after training than before. In contrast, shallow networks tend to end up with less or the same amount of pieces than at initialisation.
>
> > Generalization on more complex datasets: Beyond training accuracy, when controlling for model size and dataset size, what is the relationship between the number of pieces and test accuracy on datasets such as CIFAR100 or ImageNet?
>
> First, in all plots we report the test accuracy, not the training accuracy (we fixed the labels in the updated manuscript). We also added a new figure showing the relationship between the number of causal pieces at initialisation and final test accuracy for Fashion-MNIST (EDIT: and EuroSAT RGB, a land cover classification task using Sentinel-2 satellite images). Due to computational complexity of the model, CIFAR100 and ImageNet are currently out of scope for us, unfortunately.

---

> > ### Author Response · Authors · 2025-11-21
> >
> > We made another update to the manuscript, adding new simulation results for EuroSAT RGB (Figure 4). The EuroSAT RGB benchmark is a land cover classification task using Sentinel-2 satellite images.

---

> > > ### Comment · Reviewer_raxD · 2025-11-25
> > >
> > > Thank you for the rebuttal. I concur with the other reviewers regarding the limited theoretical depth, simple experiments, and unsurprising conclusions, and thus I maintain my original score.

---

> > > > ### Author Response · Authors · 2025-11-25
> > > >
> > > > Dear Reviewer,
> > > >
> > > > Thank you very much for your response!
> > > >
> > > > We sincerely aimed to address all of your suggestions. In particular, we have added large-scale additional experiments. Given these substantial updates and your previous remarks that were quite positive (praising the novelty, the rigorous proofs, and the applicability, e.g., "offering a principled and highly valuable metric for SNN initialization"), we are wondering if there are any remaining concerns or specific areas where the submission still falls short. We would be very grateful if you could share them. (Currently, the score sits at 4, even though the review reads more like a borderline accept.)
> > > >
> > > > Moreover, you mention the simple experiments. Can we ask in which sense the new experiments fall short of your expectations?
> > > >
> > > > The extent to which a conclusion is surprising is certainly hard to measure. Various scientists will have different perspectives. A more precise measure of the value of a conclusion could be that it is correct and scientifically rigorously argued. We believe that we have aptly demonstrated this. Would you disagree with this?
> > > >
> > > > Thank you again for your time and consideration.

---

> > > > > ### Author Response · Authors · 2025-11-28
> > > > >
> > > > > Dear Reviewer,
> > > > >
> > > > > The end of the discussion period is drawing near, and we were wondering if you had any chance to think about our questions.
> > > > >
> > > > > As far as we can tell, every reviewer question has been fully addressed, weaknesses were resolved, and strong, large-scale new experiments were included. In particular, we have now added even more experiments! We are not sure if you have already seen these.
> > > > > Yet despite very positive initial reviews, you have not posed additional questions or modified your score.
> > > > > We are interested in finding out why.

---

### Author Response · Authors · 2025-11-24

We uploaded an updated manuscript. Many more data points were added to the new EuroSAT experiment (Figure 4).

---

> ### Author Response · Authors · 2025-11-28
>
> We made a similar update for the new Fashion-MNIST experiment (Figure 4 as well).

---

### Author Response · Authors · 2025-12-02
**Brief summary for the area and program chairs**

We thank the reviewers for their thoughtful feedback and provide this brief summary for the area and program chairs.

**Contribution:**

We present a novel framework for assessing the expressivity of spiking neural networks (SNNs). Our key idea, *causal pieces*, partitions the input and parameter space into regions where the same subnetwork determines the output spikes. We prove that the number of causal pieces bounds the approximation capability of an SNN and show empirically that more pieces at initialization predict better training outcomes, making this a principled and practical tool for model analysis and improvement.

**Strengths:**
- **Solid theory and good presentation:** Described as a *“novel and natural abstraction for SNNs”* (R1) with *“rigorous and solid proofs”* (R1 & R2) and *“beautiful figures”* (R2).
- **Strong experiments:** Experiments are *“well aligned with theory”* (R2) and *“the deduction from theoretical understanding well aligns with the experimental results”* (R3).
- **Innovative aspect highlighted:** *“introduces an innovative analytical tool"* (R1).
- **Practical value highlighted:** *“a principled and highly valuable metric for SNN initialization”* (R1).

While scores do not reflect all of this, the written reviews contain **strong endorsements**.

**Clarifications:**

We addressed all weaknesses raised by the reviewers in the updated manuscript. Below, we summarize the key points.

- **Dataset complexity:** Based on reviewer requests, we added extensive results on *Fashion-MNIST* and *EuroSAT* (real satellite images), showing improved evidence (Fig. 4). These may have been missed in the discussion.
- **Neuron model:** While our main results use nLIF for analytical tractability, this is a special case of the widely used LIF model. The framework generalizes to other neuron models, since only spike times and connectivity are required to compute causal pieces.
- **Multi-spike applicability:** The framework *supports neurons that spike multiple times*, as mentioned and demonstrated in the paper.
- **“Too intuitive” concern:** We view intuitiveness as a strength - the method is broadly applicable and easy to extend.
- **“Not surprising” concern:** Surprise is not a standard evaluation criterion; we emphasize correctness, usefulness, and clarity, all of which are met in our submission.
- **Accuracy misunderstanding:** We confirmed in the updated manuscript that reported accuracies are test accuracies, not training.

We hope this summary helps clarify the value, rigor, and applicability of our work.

---

### Meta-Review · Area_Chair_1RzU · 2026-01-07

**Summary:**

This work proposes causal pieces as an analytical framework for studying the expressivity of SNNs. The paper presents a mathematically clean treatment under a specific non-leaky integrate-and-fire neuron model. It also provides empirical evidence showing correlations between the number of causal pieces at initialization and downstream performance on several benchmark tasks.

All three reviewers converged on a rejection recommendation due to the strong modeling assumptions and the lack of experiments on more complex datasets and architectures. Reviewers also noted that the main conclusions are intuitive and unsurprising, while I do not consider this to be a weakness.

Overall, I believe this is a good piece of work. I encourage the authors to consider resubmitting after establishing stronger theoretical results for more commonly used LIF neuron models and validating the framework on more complex tasks and architectures.

**Reviewer Concerns:**

During the rebuttal, most of the concerns are addressed. The authors revised the abstract, improved the introduction, clarified imprecise claims, and added additional experiments on Fashion-MNIST and EuroSAT.

However, several key concerns remain unresolved. The core theoretical results still rely on a highly simplified non-leaky integrate-and-fire neuron model, limiting the generality of the theoretical claims. Moreover, even with the added datasets, the experimental evaluation remains focused on relatively simple tasks, with comparisons primarily against basic baselines.

**Reviewer Scores:**

As concerns about limited neuron model and simple experiments tasks remains, I believe all reviewers would maintain their original scores. This is also explicitly stated by reviewers raxD and pLzs in the comments.

---

### Decision · Program_Chairs · 2026-01-26

Reject